# In vivo imaging of retrovirus infection reveals a role for Siglec-1/CD169 in multiple routes of transmission

Kelsey A Haugh[1], Mark S Ladinsky[2], Irfan Ullah[3], Helen M Stone[1], Ruoxi Pi[1], Alexandre Gilardet[1], Michael W Grunst[1], Priti Kumar[3], Pamela J Bjorkman[2], Walther Mothes[1]*, Pradeep D Uchil[1]*

[1]Department of Microbial Pathogenesis, Yale University School of Medicine, New Haven, United States; [2]Division of Biology and Biological Engineering, California Institute of Technology, Pasadena, United States; [3]Department of Internal Medicine, Section of Infectious Diseases, Yale University School of Medicine, New Haven, United States

**Abstract** Early events in retrovirus transmission are determined by interactions between incoming viruses and frontline cells near entry sites. Despite their importance for retroviral pathogenesis, very little is known about these events. We developed a bioluminescence imaging (BLI)-guided multiscale imaging approach to study these events in vivo. Engineered murine leukemia reporter viruses allowed us to monitor individual stages of retrovirus life cycle including virus particle flow, virus entry into cells, infection and spread for retroorbital, subcutaneous, and oral routes. BLI permitted temporal tracking of orally administered retroviruses along the gastrointestinal tract as they traversed the lumen through Peyer's patches to reach the draining mesenteric sac. Importantly, capture and acquisition of lymph-, blood-, and milk-borne retroviruses spanning three routes was promoted by a common host factor, the I-type lectin CD169, expressed on sentinel macrophages. These results highlight how retroviruses co-opt the immune surveillance function of tissue-resident sentinel macrophages for establishing infection.

*For correspondence:
walther.mothes@yale.edu (WM);
pradeep.uchil@yale.edu (PDU)

## Introduction

Retroviruses cause cancer and immunodeficiencies (*Blattner, 1999*). Once retroviruses establish viral reservoirs, it is difficult to eliminate infection as retroviral genomes are permanently integrated into host DNA. Despite the clinical relevance of early processes by which incoming retroviruses establish infection by navigating complex host tissue architecture *en route* to their first targets, little is known about these events (*Haase, 2010*; *Haase, 2011*; *Haase, 2014*). Retroviruses like the human immuno-deficiency virus (HIV-1) can enter through the vaginal and rectal mucosa during sexual transmission, orally via milk during mother-to-child transmission, and subcutaneously and intravenously through needle stick injections during drug use and blood transfusions (*Haase, 2010*; *Friedland and Klein, 1987*). Most murine leukemia virus (MLV) transmission in mice occurs vertically from dam-to-pup via ingestion of virus-containing milk through the gastrointestinal (GI) tract. MLV transmission can also occur parenterally between male mice during infighting and via the venereal route between infected male and female mice (*Portis et al., 1987*; *Buffett et al., 1969a*). Entry via different routes requires retroviruses to navigate diverse host tissue architecture and overcome barriers for successful infec-tion (*Uchil et al., 2019a*; *Pfeiffer, 2010*; *Fackler et al., 2014*). Whether retroviruses exploit common host factors across these transmission routes remain to be clarified.

We have previously used MLV as a model retrovirus to understand how retroviruses establish infection in mice through the lymph or blood following subcutaneous and intravenous delivery,

respectively (*Sewald et al., 2015*; *Uchil et al., 2019b*). We found that sentinel macrophages lining blood/lymph-tissue interfaces such as the subcapsular sinus in lymph nodes or the marginal zones in the spleen function to filter out incoming retroviruses from *Sewald et al., 2015*; *Uchil et al., 2019b*. The 'fly-paper'-like activity of sentinel macrophages has been observed for various incoming viruses and pathogens (*Kastenmüller et al., 2012*; *Iannacone et al., 2010*; *Junt et al., 2007*). The frontline position of sentinel macrophages allows them to orchestrate downstream innate, cell-mediated, and humoral immune responses to incoming pathogens in the lymph and blood (*Uchil et al., 2019a*; *Uchil et al., 2019b*; *Honke et al., 2012*; *Martinez-Pomares and Gordon, 2012*; *Perez et al., 2017*). These macrophages naturally express the I-type lectin Siglec-1/CD169, which specifically interacts with sialosides present on retroviral membranes (*Sewald et al., 2015*; *Izquierdo-Useros et al., 2012*; *Puryear et al., 2013*). CD169 expression allows sentinel macrophages to capture retroviruses and limits their dissemination (*Uchil et al., 2019b*). However, retroviruses like MLV and HIV-1 exploit their CD169 to promote infection of target lymphocytes that sample antigens captured by sentinel macrophages (*Sewald et al., 2015*; *Uchil et al., 2019b*; *Pi et al., 2019*). Whether the observed roles for CD169$^+$ macrophages following subcutaneous and intravenous delivery are of any relevance for natural mother-to-offspring transmission when viruses enter via the GI tract has remained unknown.

Identifying portals of entry such as Peyer's patches (PP) in underdeveloped intestines can be very challenging, given the complexity, functional diversity, and length of the GI tract, which can measure >7 cm even in neonatal mice (*Torow et al., 2017*). Here, we developed a whole-body bioluminescence imaging (BLI)-based approach to illuminate areas where MLV concentrates to traverse into the gut tissue from the lumen. We implemented BLI by developing a series of MLV-based reporter viruses to enable observation of specific stages of the retrovirus life cycle in vivo, including viral particle flow, entry into the cytoplasm, first infection events, and spread. We first validated this system by testing its ability to uncover new insights into previously studied subcutaneous and intravenous transmission routes (*Sewald et al., 2015*; *Uchil et al., 2019b*; *Pi et al., 2019*). Second, BLI permitted temporal tracking of various steps of virus infection for orally administered retroviruses along the GI tract as they traversed the lumen through the PP to reach the draining mesenteric sac (mSac). Finally, we show that capture and acquisition of lymph-, blood-, and milk-borne retroviruses spanning three routes was promoted by a common host factor, CD169, expressed on sentinel macrophages. Our results highlight how retroviruses co-opt the immune surveillance function of tissue-resident sentinel macrophages for establishing infection. Understanding these events will inform design of improved prophylactic strategies that target prevention of virus acquisition and establishment of infection.

## Results

### Generation of reporter viruses to enable visualization of individual stages of retrovirus infection in vivo

We established a BLI-directed approach for studying individual stages of retroviral infection in vivo by strategically inserting reporters into unique sites in the MLV genome (*Figure 1A*). To track virus particle flow, we generated bioluminescent virus particles by introducing nanoluciferase (Nluc) into the proline-rich region (PRR) of the MLV envelope (Env). Nluc-Env-tagged virus particles produced using a tripartite plasmid system (encoding Nluc-tagged MLV Env, MLV Gag-Pol, and MLV-LTR) exhibited 200 times more luciferase activity per virus particle (0.2 RLU/virion) compared to viruses generated using the full-length MLV genome (0.001 RLU/virion) (*Figure 1B*). To monitor virus entry into the cytoplasm of cells, we fused firefly luciferase (Fluc) to the C-terminus of MLV Gag (MLV Gag-Fluc Env$_{WT}$) and exploited the ATP-dependence of Fluc activity, which is restricted to the host cell cytoplasm in vivo. This strategy ensured that Fluc activity was exhibited when both detergent (Triton X-100) and ATP were present (*Figure 1C*). A Gag-Fluc-labeled virus in which MLV envelope was replaced by the fusion-defective SFFV gp55 envelope (MLV Gag-Fluc Env$_{FD}$) served as a negative control. To monitor single-round virus transduction, we utilized a replication-defective virus generated by co-transfecting a dual BLI and GFP reporter (pMIG-Nluc-IRES-GFP) in conjunction with MLV Gag-Pol and Env (*Ventura et al., 2019*). In addition, we generated red-shifted reporter viruses encoding Antares, which is a bioluminescence resonance energy transfer (BRET) reporter that enables superior deep-tissue sensitivity over Nluc in vivo (*Chu et al., 2016*). Finally, to permit

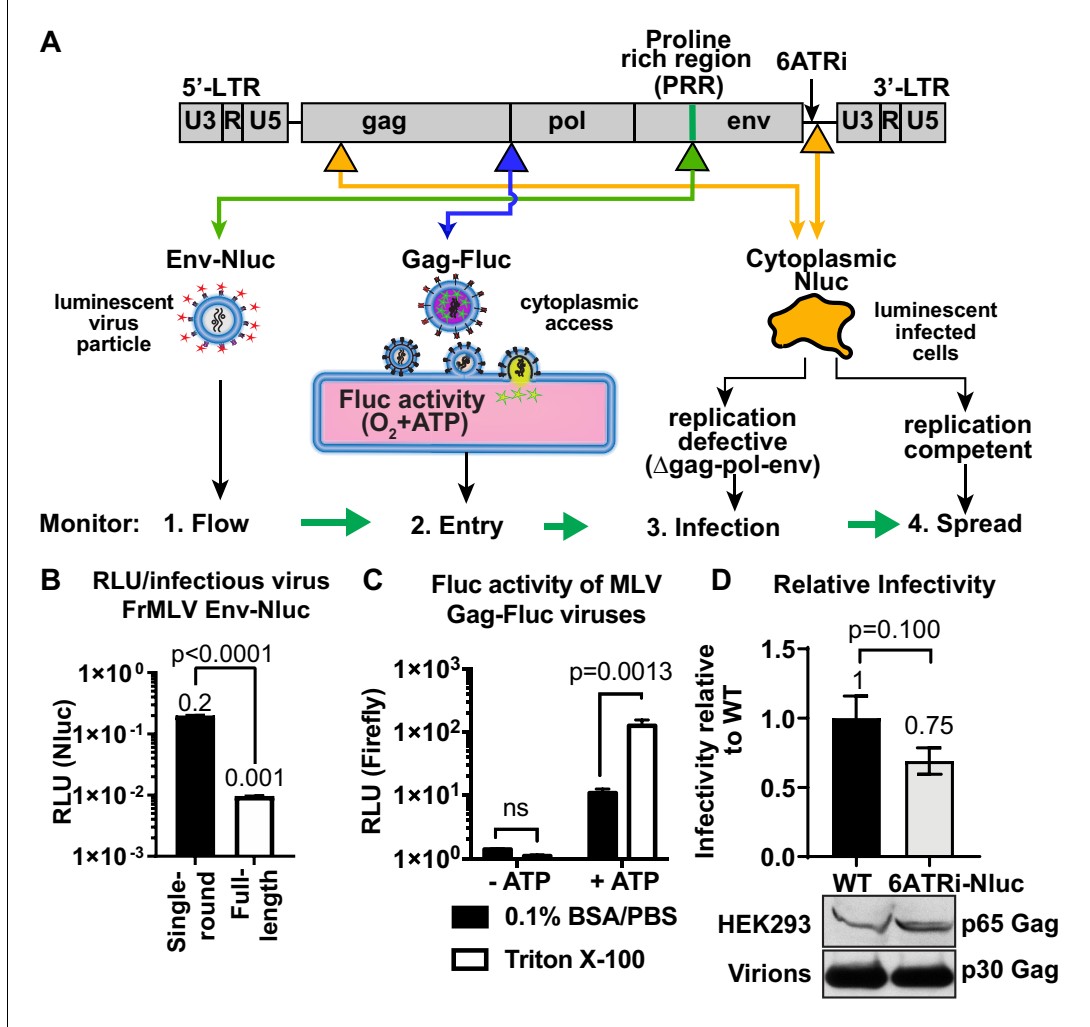

**Figure 1.** Construction and characterization of reporter viruses for visualizing individual stages of the retrovirus life cycle in vivo. (A) A scheme denoting the location of inserted reporters into unique sites in the Friend murine leukemia virus (FrMLV) genome. (1) To monitor particle flow, Nluc was inserted in-frame into the proline-rich region (PRR) of envelope. (2) To monitor virus entry into the cytoplasm of cells, Gag-Firefly luciferase (MLV Gag-Fluc; ΔPro-Pol) was employed as Fluc requires both oxygen and ATP in the presence of substrate D-luciferin for its activity. (3) To monitor infection, Nluc was expressed in the cytoplasm using a viral LTR-driven construct (single round; Δgag-pol-env). (4) To monitor spreading infection, Nluc was expressed from a short internal ribosome entry site (IRES) element (6ATRi) downstream of the envelope gene that resulted in replication-competent virus. (B) A graph comparing brightness (relative light units [RLU]) per infectious unit of full-length FrMLV Env-Nluc viruses with single-round MLV generated using gag-pol, env-Nluc, and LTR-GFP constructs. Infectious units for both viruses were estimated using DFJ8 cells followed by flow cytometry. RLU per infectious virion was determined by measuring Nluc activity in sedimented virus. The error bars denote standard deviations between triplicate samples. p values derived from Student's t-test. (C) A graph showing detectable Fluc activity in intact (0.1% BSA/PBS) or lysed (0.1% BSA/PBS, Triton X-100) MLV Gag-Fluc virions in the presence or absence of ATP and substrate D-luciferin (15 mg/mL in PBS). The error bars denote standard deviations between triplicate samples. p values derived from Student's t-test; ns: not significant. (D) A graph comparing released infectivity of replication-competent FrMLV Nluc reporter virus (6ATRi-Nluc) and WT FrMLV (WT). Viruses were produced by transfecting equal amounts of virus-encoding plasmids into HEK293 cells in triplicate. 48 hr post-transfection released infectivity in culture supernatants was determined using DFJ8 target cells followed by flow cytometry with antibodies to MLV GlycoGag to enumerate infected cells. Released infectivity of WT FrMLV was set to 1. The error bars denote standard deviations between triplicate samples. Western blot analyses of sedimented virus from culture supernatants and HEK293 cell lysates for a similar experiment as shown in the graph above using antibodies to MLV Gag.

longitudinal monitoring of progressing infection, we generated replication-competent MLV reporter viruses by introducing a shortened internal ribosome entry site (IRES), 6ATRi, to drive Nluc expression downstream of viral Env (MLV 6ATRi-Nluc) (*Ventura et al., 2019*; *Alberti et al., 2015*; *Logg et al., 2001*). This strategy enabled bi-cistronic expression of Env and Nluc in infected cells.

Infectivity measurements revealed that MLV 6ATRi-Nluc was ~60% as infectious as wild-type MLV (*Figure 1D*).

## BLI-driven characterization of blood-borne retrovirus infection

MLV infection via the intravenous route (retroorbital [r.o.]) is a well-studied infection route. We previously showed that r.o.-delivered MLV is first captured by CD169[+] metallophilic marginal zone macrophages before infecting follicular, marginal zone, and transitional B cells (*Uchil et al., 2019b*). We revisited this route at the whole-body level by applying these new reporter viruses. We challenged mice r.o. with MLV Env-Nluc reporter viruses. BLI-driven virus tracking immediately after challenge showed that MLV rapidly reached both liver and spleen within 30 s after administration and accumulated predominantly at the spleen after passing through the liver (2–3 min) (*Figure 2A, B*, *Figure 2— video 1*). This was consistent with the spleen being the main blood-filtering organ in mice. Interestingly, the decaying luminescence over time (>30 min) in the spleen was revived when Nluc substrate (furimazine) was readministered (*Figure 2—figure supplement 1*). This indicated that viruses remain captured at the spleen, and the exhaustion of substrate contributed to the decay in the signal. We next investigated virus entry into the host cell cytoplasm by utilizing MLV Gag-Fluc-tagged viruses with wild-type (Env$_{WT}$) or fusion-defective Env (Env$_{FD}$). Inocula were equalized by measuring the luciferase activity (relative light units [RLU]) in detergent-lysed viral preparations (*Figure 2C*). Mice challenged r.o. with MLV were monitored at 5 min, 30 min, and 1 hr post infection (hpi) using BLI. In contrast to animals infected with fusion-defective MLV, we observed Fluc signal emerging at the spleen (1 hpi) in animals infected with reporter viruses carrying wild-type MLV envelope (*Figure 2D, E*; p=0.0061). Taken together, our data indicated that blood-borne MLV was filtered rapidly at the spleen within 2–3 min and entered the cytoplasm of cells by 60 min after capture.

We next infected mice with WT FrMLV or MLV 6ATRi-Nluc reporter virus and compared infection levels at 7 days post infection (dpi). In vivo infectivity of MLV 6ATRi-Nluc virus was reduced in comparison to WT FrMLV (*Figure 2F*). This was not unexpected based on the reduced released infectivity of MLV 6ATRi-Nluc in vitro (*Figure 1D*), and the known effect of genomic reporter insertions on retrovirus fitness (*Logg et al., 2001*). To visualize the first round of infected cells and virus spread at the whole animal level, we challenged mice with single-round MLV reporter virus (pMIG-Antares) as well as replication-competent reporter MLV and monitored replication dynamics using BLI every 2–3 days over the course of 2 weeks (*Figure 2G*). In contrast to the decline of luminescent signal observed with single-round Antares-encoding MLV, luminescent signal in organs infected by MLV 6ATRi-Nluc increased over time, indicative of fresh rounds of infection (*Figure 2G, H*). We observed infection in auricular and inguinal LN in addition to the spleen and liver. As time progressed, there was gradual decline of luminescent signal in mice infected with MLV 6ATRi-Nluc (*Figure 2G, H*). This decline was not due to loss of the Nluc reporter as RT[2]-qPCR analyses of viral RNA isolated from blood and spleen indicated that the ratio of Gag to Nluc did not significantly change during the observed course (3–14 dpi) of infection in mice (*Figure 2I*). The Gag:Nluc ratio was similar to that seen from RNA isolated from DFJ8-infected cells (30 hpi). These data suggested that Nluc reporter was retained within the viral genome throughout the observed course of infection. Therefore, the decline in the signal is consistent with the immune control of MLV infection in C57BL/6J (B6) mice, observed in previous studies (*Sewald et al., 2015*; *Uchil et al., 2019b*; *Pi et al., 2019*; *Sewald et al., 2012*; *Nowinski, 1976*). Taken together, these results demonstrate the utility of our bioluminescent reporter viruses in monitoring particle flow, capture, cytoplasmic entry, transduction, and subsequent virus spread following intravenous infection of mice. The validation of this reporter system also set the stage for applications to other infection routes.

## BLI-driven characterization of lymph-borne retrovirus infection

Intrafootpad (i.f.p.) infection is widely used to study subcutaneous (s.c.) infection and to model antigen trafficking to draining lymph nodes via lymphatics (*Sewald et al., 2016*; *Chatziandreou et al., 2017*). The draining lymph node for i.f.p. infection is the popliteal lymph node (pLN) (*Iannacone et al., 2010*; *Junt et al., 2007*; *Gonzalez et al., 2010*). Previous studies using multi-photon intravital microscopy indicated that incoming lymph-borne viruses were captured by CD169[+] sentinel macrophages, resulting in accumulation at the subcapsular sinus of the pLN within a few minutes following viral challenge (*Sewald et al., 2015*; *Iannacone et al., 2010*; *Junt et al., 2007*;

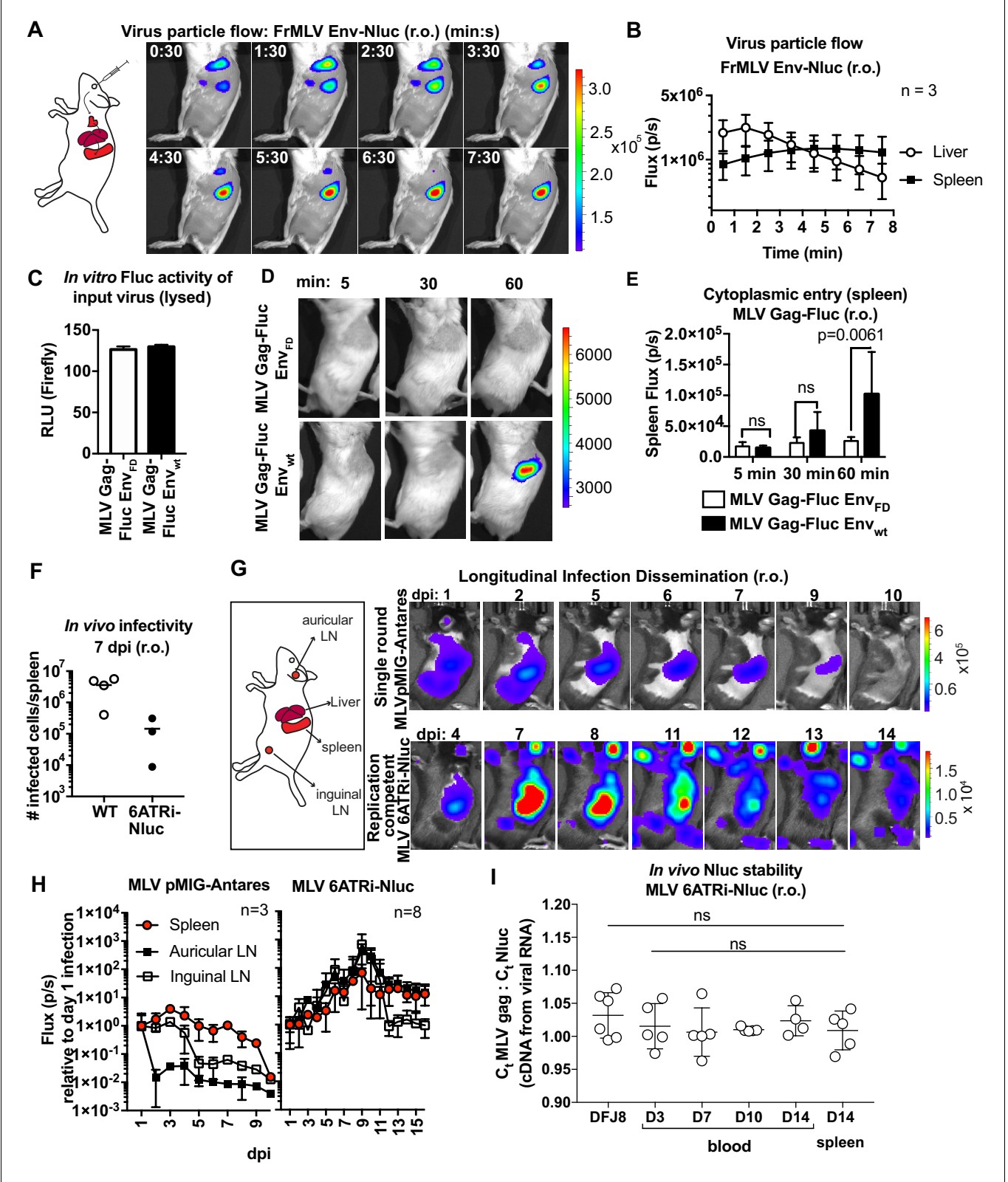

**Figure 2.** Real-time visualization of individual steps of retrovirus infection in vivo during retroorbital challenge. (A) A scheme showing the path of murine leukemia virus (MLV) Env-Nluc particles after intravenous retroorbital (r.o.) challenge. Furimazine (Nluc substrate)-administered mice were challenged retroorbitally with $1 \times 10^5$ IU of MLV Env-Nluc and monitored using IVIS at 30 s (s) intervals. Images from one representative experiment from three biological replicates (n = 3) are shown. (B) Quantification of MLV Env-Nluc bioluminescent signal in the spleen and liver, displayed as photon

*Figure 2 continued on next page*

*Figure 2 continued*

flux (photons/s) in each organ following r.o. challenge from the experiment described in (A). Curves represent mean flux over time, and error bars denote standard deviation. (C) A plot showing Fluc activity (relative light units [RLU]) associated with MLV Gag-Fluc Env$_{WT}$ or MLV Gag-FLuc Env$_{FD}$ (fusion defective) inocula after lysis. Mice were subsequently inoculated with $2 \times 10^4$ RLU of indicated unlysed virus preparations. Error bars denote standard deviations from mean. (D) Representative images of MLV Gag-Fluc Env$_{WT}$ entry into the target cell cytoplasm at the spleen observed via BLI at the indicated time points following r.o. infection. Mice challenged similarly with MLV Gag-Fluc Env$_{FD}$ served as negative controls for determination of background signals. Images from one representative experiment from 4 to 7 biological replicates are shown. (E) Quantification of MLV cytoplasmic entry in the spleens (photon flux/s) of mice (n = 4–7) following r.o. infection with MLV Gag-Fluc Env$_{WT}$ or MLV Gag-Fluc Env$_{FD}$ at indicated time points from the experiment as in (D). p values derived from non-parametric Mann–Whitney test; ns: not significant; bars represent mean values and error bars denote standard deviations from mean. (F) A plot showing total number of infected cells per spleen of B6 mice (n = 3–4) challenged with $5 \times 10^5$ IU of FrMLV (WT) and replication-competent reporter FrMLV (MLV 6ATRi-Nluc) at 7 days post infection (dpi) (r.o.). Infected cells in single-cell suspensions of spleen were determined using flow cytometry with antibodies to MLV GlycoGag. Horizontal lines represent mean values. (G) Mice were infected r.o. with $5 \times 10^5$ IU of replication-competent reporter FrMLV (MLV 6ATRi-Nluc or single-round MLV pMIG-Antares) as a control for tracking dissemination of transduced infected cells. Dissemination of viral infection was monitored via bioluminescence imaging (BLI) at the indicated time points. Infected organs are indicated in the schematic. Images from one representative experiment are shown. (H) Quantification of virus dissemination in indicated organs is displayed as photon flux in mice infected with MLV 6ATRi-Nluc or MLV pMIG-Antares for the experiment described in (G). Antares, n = 3; 6ATRi-Nluc, n = 8. Images in (G) are from one representative experiment. Symbols represent mean values, and error bars denote standard deviations from mean. Scale bars that accompany the images showing luminescence denote radiance in photons per second per square centimeter per steradian (p/s/cm$^2$/sr). (I) In vivo longitudinal stability of Nluc within the MLV 6ATRi-Nluc genome was determined via RT$^2$-qPCR in B6 mice. RNA extracted from blood or spleen of mice (n = 5) r.o. infected with MLV 6ATRi-Nluc ($2 \times 10^6$ IU) at the indicated time points. Infected spleens were harvested at 14 dpi. RNA from DFJ8-infected cells (MLV 6ATRi-Nluc; $2 \times 10^5$ IU) and uninfected cells served as positive and negative controls for PCR. RT$^2$-qPCR was performed on viral RNA extracted from infected samples using primers for Nluc and MLV gag. Stability of Nluc within the viral genome was determined by ratios of gag:Nluc C$_t$ values over time.

The online version of this article includes the following video and figure supplement(s) for figure 2:

**Figure supplement 1.** Stable retention of murine leukemia virus (MLV) Env-Nluc in the spleen demonstrated by readministration of furimazine substrate after retroorbital (r.o.) challenge.

**Figure 2—video 1.** Murine leukemia virus (MLV) particle flow through the liver and spleen in the retroorbital infection route.

https://elifesciences.org/articles/64179#fig2video1

---

*Murooka et al., 2012*). CD169$^+$ macrophages did not get infected but rather promoted *trans*-infection of susceptible lymphocytes such as innate-like B1-a cells and CD4$^+$ T cells during the first round of infection. B-1a cells then spread the virus to naïve B-2 cells via virological synapse during the expansion phase of the infection (*Pi et al., 2019*; *Sewald et al., 2015*). To gain further insight into behaviors of incoming virus particles in this well-studied route, we performed BLI imaging of incoming virus particle flow from the administration site to the target organ by infecting mice i.f.p. with MLV Env-Nluc. Incoming viruses accumulated rapidly at the pLN, with detectable signal occurring within 1 min 30 s pi (*Figure 3A, B*, *Figure 3—video 1*). This was indicative of lymph flow-mediated dissemination of MLV to pLN and was consistent with previous multiphoton microscopy studies (*Sewald et al., 2015*). However, we observed that most of the incoming virus particles localized to the injection site at the footpad (*Figure 3A, B*). Quantification of virus particle accumulation in the footpad and pLN, displayed as photon flux (photons/s), revealed that virus accumulation in the footpad was over 200-fold higher than that in the pLN. This observation remained constant for the entire imaging window. Even when virus eventually accumulated in the pLN, plateauing at ~9 min pi, the level of virus in the footpad remained ~150-fold above that of the pLN (*Figure 3A, B*). Thus, our imaging analyses showed that only a small fraction of incoming viruses surpass tissue barriers including collagen fibrils, muscle tissue, and antigen-capturing cells at the footpad to reach the draining site.

We next explored the location where viruses fused and gained access to the host cell cytoplasm. We challenged mice with MLV Gag-Fluc Env$_{WT}$ or MLV Gag-Fluc Env$_{FD}$ via the i.f.p. route and monitored these events by BLI. Bioluminescent signal in the mouse footpads infected with MLV Gag-Fluc Env$_{FD}$ minimally increased over time (*Figure 3C, D*). In contrast, luciferase activity in footpads challenged with MLV Gag-Fluc Env$_{WT}$ significantly increased over time, indicating progressive virus access to cell cytoplasm in vivo (*Figure 3C, D*). We observed that fusion-competent viruses gained access to the host cell cytoplasm in the footpad as early as 3 min post-challenge (*Figure 3C, D*, *Figure 3—figure supplement 1A, B*). Despite accumulation of virions in the pLN within minutes following challenge (*Figure 3A, B*), we were unable to detect Fluc activity in the first 40 min of the

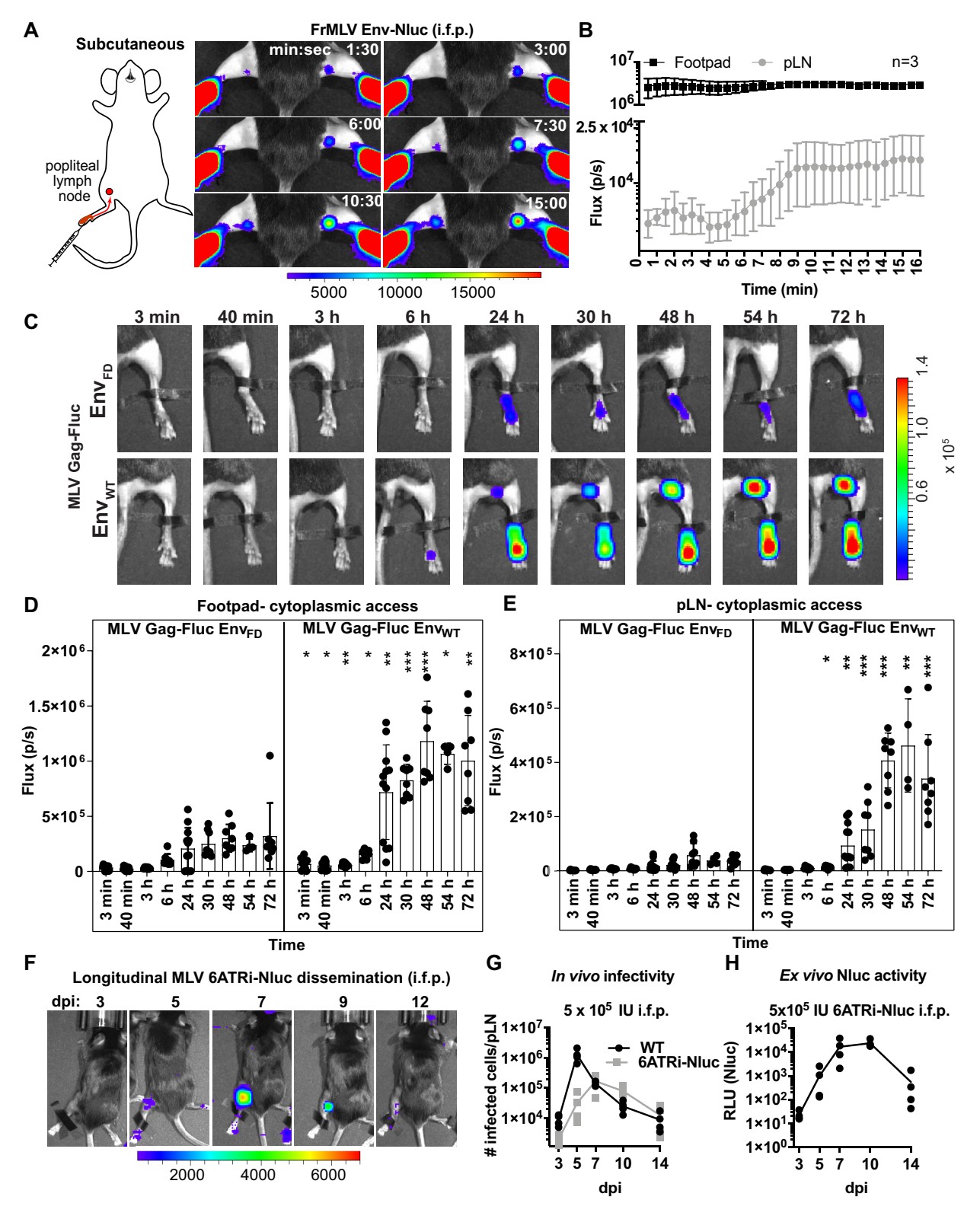

**Figure 3.** Real-time visualization of individual steps of retrovirus infection in vivo during subcutaneous challenge. (**A**) A scheme showing the path of murine leukemia virus (MLV) Env-Nluc from footpad to draining popliteal lymph node (pLN) after subcutaneous (intrafootpad [i.f.p.]) challenge in mice. Furimazine (Nluc substrate)-administered mice were challenged i.f.p. with $1 \times 10^5$ IU of MLV Env-Nluc and monitored using IVIS at 30 s (s) intervals. Images from one representative experiment from three biological replicates (n = 3) are shown. (**B**) Quantification of MLV accumulation in footpad and

*Figure 3 continued on next page*

Figure 3 continued

pLN, displayed as photon flux (photons/s) at each site following i.f.p. inoculation with MLV Env-Nluc (n = 3), from the experiment described in (A). Curves represent mean flux over time, and error bars denote standard error from mean. (C) Representative images of MLV Gag-Fluc Env$_{WT}$ entry into the target cell cytoplasm at the footpad and pLN observed by bioluminescence imaging (BLI) at the indicated time points following i.f.p. infection. Mice challenged similarly with MLV Gag-Fluc Env$_{FD}$ (fusion-defective envelope) served as negative controls for determination of background signals. Fluc activity was measured for both virus preparations after detergent lysis and equivalent amounts ($2 \times 10^6$ relative light units [RLU]) were delivered i.f.p. before imaging using IVIS. Images from one representative experiment from 8 to 10 biological replicates are shown. (D, E) Quantification of MLV cytoplasmic entry in footpads (D) and pLNs (E) (photon flux/s) of mice (n = 8–10) following i.f.p. infection with MLV Gag-Fluc Env$_{WT}$ or MLV Gag-Fluc Env$_{FD}$ at indicated time points from the experiment as in (C). p values derived from non-parametric Mann–Whitney test of MLV Env$_{WT}$- vs. MLV Env$_{FD}$-infected mice; *p<0.05; **p<0.01; ***p<0.001; ****p<0.0001; mean values ± SD are depicted. (F) Mice were infected with $4 \times 10^5$ IUs of replication-competent reporter FrMLV (MLV 6ATRi-Nluc [i.f.p.]). Dissemination of viral infection was monitored via BLI at the indicated time points days post infection (dpi). Images from one representative experiment are shown. (G) A plot showing total number of infected cells per pLNs of B6 mice (n = 3–4) challenged with $4 \times 10^5$ IU of FrMLV (WT) (i.f.p.) and replication-competent reporter FrMLV (MLV 6ATRi-Nluc) at indicated days post infection (dpi). Infected cells in single-cell suspensions of pLNs were determined using flow cytometry with antibodies to MLV GlycoGag. Connecting lines representing mean values along with individual data points are shown. (H) A plot showing ex vivo Nluc activity in cells isolated from pLNs of mice infected with MLV 6ATRi-Nluc-infected from the experiment shown in (F). Individual data points along with connecting lines (mean values) are shown. Scale bars that accompany the images denote radiance in photons per second per square centimeter per steradian (p/s/cm$^2$/sr).

The online version of this article includes the following video and figure supplement(s) for figure 3:

**Figure supplement 1.** Characterization of murine leukemia virus (MLV) infection during footpad challenge.

**Figure 3—video 1.** Murine leukemia virus (MLV) particle flow to the popliteal lymph node following intrafootpad (i.f.p.) inoculation.

https://elifesciences.org/articles/64179#fig3video1

continuous imaging time frame (*Figure 3C, E*). However, there was a significant increase in the Fluc signal at the pLN with viruses carrying Env$_{WT}$ compared to Env$_{FD}$, suggesting cytoplasmic access as early as 6 hpi that gradually increased thereafter (*Figure 3E*, *Figure 4—figure supplement 1A, B*). The delay was likely due to the mode of infection at the pLN, where MLV is first captured by CD169 macrophages before *trans*-infection of permissive B-1a cells (*Pi et al., 2019*; *Sewald et al., 2012*).

Next, we infected mice subcutaneously (s.c.) in the footpads with MLV 6ATRi-Nluc or WT MLV, harvested pLN at 3, 5, 7, 9, or 11 dpi, and assessed the number of infected cells in individual pLNs by flow cytometry using antibodies to MLV GlycoGag. WT MLV exhibited an expected infection profile in vivo, peaking at 5 dpi and subsequently decreasing due to immune control (*Figure 3F, G*). As in r.o. infection (*Figure 2F–H*), MLV 6ATRi-Nluc replicated with slower infection kinetics compared to those of WT MLV and peaked at 7–8 dpi instead of 5 dpi (*Figure 3F, G*). We also confirmed luciferase activity resulting from single-cell suspensions isolated from pLN isolated from mice infected with MLV 6ATRi-Nluc at different time points (*Figure 3H*). Luciferase activity mirrored infection curves obtained by flow cytometric enumeration of infected cells using antibodies to MLV GlycoGag. In contrast to r.o. challenge, virus infection during s.c. challenge was restricted to the draining pLN and new infection events were not observed beyond the target organ before elimination by mounting immune responses in resistant B6 mice (*Figure 3F–H*). These data are consistent with our earlier work describing how capture by CD169$^+$ macrophages at pLN limits systemic spread and initiates effective immune responses (*Uchil et al., 2019b*; *van Dinther et al., 2018*).

## CD169$^+$ macrophages contribute to virus capture in the footpad

We next characterized the cell types that capture MLV at the footpad by challenging mice with MLV Gag-GFP particles (i.f.p.) (*Figure 4A*). Surprisingly, immunostaining of footpad cryosections (15 min post-challenge) revealed that in addition to CD11c$^+$ dendritic cells (DCs), CD169$^+$ macrophages also reside in the footpad and predominantly captured MLV (*Figure 4A*). These results are consistent with our previous studies in the pLN demonstrating that CD169$^+$ macrophages capture incoming virus particles (*Uchil et al., 2019b*; *Pi et al., 2019*; *Sewald et al., 2015*). Electron tomography revealed the presence of virus-laden macrophages with viruses present in membrane invaginations, as well as tethered to plasma membranes, suggestive of CD169-mediated capture (*Figure 4A*, *Figure 4—figure supplement 1*, *Figure 4—video 1*). Quantification of cell-associated viruses in tomographic sections was consistent with immunostaining data (*Figure 4A*) and revealed that incoming viruses predominantly associated with macrophages (*Figure 4*, *Figure 4—figure supplement 1*) and to a lesser extent with DCs. In many cases, we observed virus-capturing CD169$^+$ macrophages in

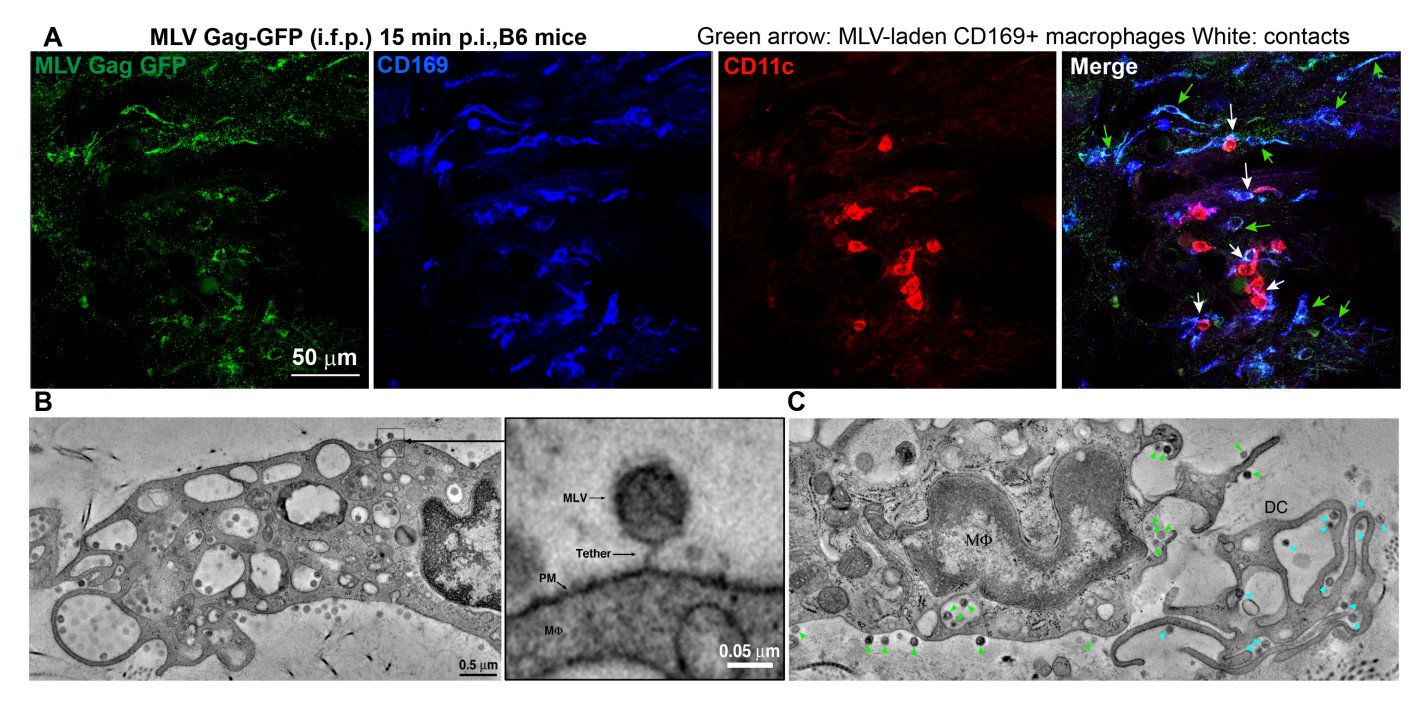

**Figure 4.** Sentinel macrophages mediate capture of incoming viruses at the injection site during subcutaneous challenge. (**A**) Images of footpad cryosections from B6 mice 15 min after intrafootpad (i.f.p.) administration of murine leukemia virus (MLV) Gag-GFP virus particles (green). Macrophages and dendritic cells (DCs) were identified using antibodies to CD169 (blue) and CD11c (red), respectively. Green arrows denote MLV-laden CD169[+] macrophages, and white arrows denote contacts between CD169[+] macrophages and DCs. (**B**) Electron tomography of footpads from B6 mice prepared 15 min post-challenge with WT FrMLV. MLV virions can be seen associated with plasma membranes (PM) invaginations of macrophages (mΦ) as well as via a presumed CD169 tether in the magnified inset. (**C**) Electron tomography showing synaptic, MLV-containing contacts between macrophages and DCs in footpads of B6 mice at 15 min post-challenge (i.f.p.) WT FrMLV. Green arrowheads indicate macrophage-associated MLV particles, and blue arrowheads indicate DC-associated MLV particles. Scale bars as indicated.

The online version of this article includes the following video and figure supplement(s) for figure 4:

**Figure supplement 1.** Murine leukemia virus (MLV) primarily associates with sentinel macrophages in the footpad.

**Figure 4—video 1.** Murine leukemia virus (MLV) is captured by footpad macrophages following subcutaneous virus challenge.
https://elifesciences.org/articles/64179#fig4video1

close contact with DCs, indicative of synaptic cell-cell contacts (*Figure 4A*). We also observed viruses within cell-cell contacts between virus-capturing macrophages and DCs (*Figure 4B, C*). Indeed, characterization of cells isolated from footpad (3 dpi) by flow cytometry revealed DCs (CD11c[+] CD11b[+], and CD11c[+]) were the predominant cell types that became infected by MLV (*Figure 3—figure supplement 1C, D*).

## BLI-driven characterization of oral route of retrovirus transmission

We utilized the capacity of BLI to pinpoint events of interest in extensive organs like the GI tract by characterizing individual steps in the less understood oral route of MLV transmission. To study mother-to-offspring transmission, we infected a lactating dam in the mammary glands (s.c.) with WT MLV carrying a co-packaged Fluc reporter driven from the viral LTR and allowed the infection to establish for 6 days. Longitudinal BLI confirmed the presence of luminescent signal in infected teats at 6 dpi (*Figure 5A*, *Figure 5—figure supplement 1*). Immunohistochemistry of luminescent mammary glands at 7 dpi revealed FrMLV-infected epithelial cells and virus particles in mammary gland tissue (*Figure 5C*). Interestingly, we noted that the progression of infection in mammary glands was asynchronous (*Figure 5B*, *Figure 5—figure supplement 1*). As MLV requires cell division for infection, the asynchronous viral replication may reflect different levels of cell division in mammary glands during lactation. Subsequent transfer of neonatal mice (1–3 days old) for fostering resulted in successful transmission of MLV as seen by luciferase-positivity in the GI regions at 2 days post-transfer

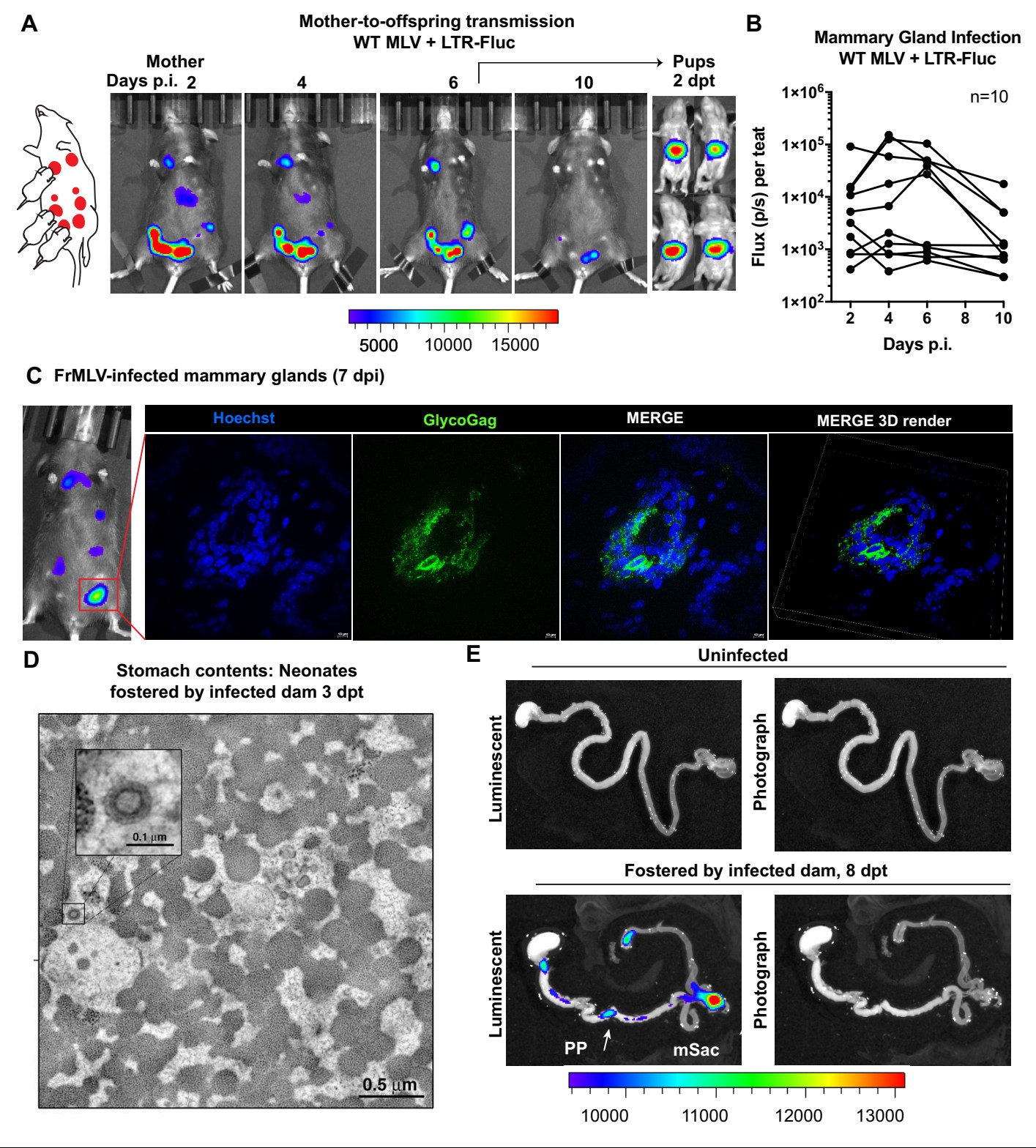

**Figure 5.** Visualization of murine leukemia virus (MLV) transmission from mother to offspring. (**A**) A scheme showing MLV-infected dam fostering pups. $1 \times 10^7$ IU of WT FrMLV carrying MLV LTR-Fluc were inoculated by distribution in mammary glands of a lactating dam. Virus replication in the dam was monitored longitudinally via NBLI. At 6 days post infection (dpi), neonatal mice from a separate litter were provided to foster and successful MLV transmission from dam to pups was visualized 2 days post-transfer (dpt) by NBLI. (**B**) Fluc activity indicating virus replication in infected teats was measured longitudinally and plotted as photon flux (p/s); flux from individual teats is shown. (**C**) Immunostained images of cryosections obtained from

*Figure 5 continued on next page*

*Figure 5 continued*

indicated mammary gland of a lactating dam infected as in (**A**) (7 dpi). MLV-infected epithelial cells (green) in the teats were identified using antibodies to MLV GlycoGag and nuclei were stained using Hoechst 3342 (blue). 3-D volume-rendered images were generated from z-planes images taken 0.3 μm apart. (**D**) Electron tomography of stomach contents at 3 dpt from an infected neonate for an experiment as in (**A**), revealing free viruses in the stomach. Inset: Details of a single cell-free MLV particle within the sample volume. Scale bars as indicated. (**E**) Merged luminescence and photographs gastrointestinal (GI) tract from an uninfected neonatal mouse or from a neonatal mouse that was allowed to feed for 8 days (8 dpt) from infected dam for an experiment as in (**A**) showing luminescent signal in Peyer's patches (PP) and mesenteric LN (mLN). Scale bars shown for bioluminescence imaging denote radiance in photons per second per square centimeter per steradian (p/s/cm$^2$/sr).

The online version of this article includes the following figure supplement(s) for figure 5:

**Figure supplement 1.** Murine leukemia virus (MLV) replication in mammary glands of infected, lactating dams.

(dpt) (*Figure 5A*). Electron tomography of neonatal stomach contents at 3 dpt revealed cell-free viruses in milk (*Figure 5B*), congruent with studies of mother-to-offspring transmission of cell-free MMTV to suckling pups (*Ross, 2000*; *Courreges et al., 2007*). Due to the internal location and convoluted nature of the GI tract, necropsy was required for revealing anatomic details of the infected regions. BLI analyses of GI tracts from neonates fostered by infected dams revealed that viruses had established infection in the PP and mSacs by 8 dpt (*Figure 5D*).

To explore the dynamics of virus transit in the intestinal tract, we orally inoculated 3-day-old mice with replication-defective MLV Env-Nluc particles and analyzed their distribution at 12, 24, or 48 hpi via BLI. We observed a sequential movement of virus particles from the stomach at 12 hpi to a striking, temporal accumulation in PP and mSacs of the small intestine (*Figure 6A, B*). There was a significant and simultaneous increase in particle accumulation between 24 and 48 hpi at both PP and mSac (*Figure 6B*). These results indicated that incoming particles from PP can reach the draining mSac via the lymph drainage without first undergoing replication. There was less frequent accumulation of incoming particles in cecal patches, likely because they are downstream of PP in the direction of intestinal traffic. Our data reveal that PP, followed by mSacs, were the earliest intestinal structures to accumulate incoming MLV.

To determine the tissue sites where incoming viruses entered the cytoplasm of intestinal target cells, we orally infected neonatal mice with MLV Gag-Fluc Env$_{WT}$ or MLV Gag-Fluc Env$_{FD}$, which served as a control. Fluc signal was first observed in PP (*Figure 6C*). We saw a significant temporal increase in access to the cytoplasm of target cells of PPs by incoming MLV Gag-Fluc Env$_{WT}$ viruses compared to the fusion-deficient MLV Gag-Fluc Env$_{FD}$ control viruses (*Figure 6D*). However, Fluc activity at the mSac began to increase weakly over control particles only by 48 hpi (*Figure 6C, E*), despite early particle accumulation (*Figure 6B*). Delayed cytoplasmic access was a recurring theme, suggesting that antigen-presenting cell-mediated *trans*-infection processes, as seen in pLN, are also effective in mesenteric LNs. Finally, the use of single-round (MLV Nluc-IRES-GFP) and replication-competent viruses (MLV 6ATRi-Nluc) confirmed true virus infection events at both PP and mSacs at 96 hpi by BLI (*Figure 6F, G*). Infection levels, measured by flux, were expectedly lower in PPs compared to the draining mSacs, which accumulated more viruses than PP (*Figure 6B, G*). A characterization of cells isolated from neonatal mSacs at 5 dpi revealed that infected cells were predominantly CD11c$^+$ DCs (~50%), followed by CD19$^+$ B cells (~35%) and CD4$^+$ T cells (~15%) (*Figure 6—figure supplement 1*).

## Incoming retroviruses enter Peyer's patches through M cells

Microfold (M) cells in PP function as portals of entry into underlying lymphoid follicles for particulate antigens, bacteria, and viruses such as MMTV present in the lumen (*Ohno, 2016*; *Miller et al., 2007*; *Buffett et al., 1969b*). M cells have been previously implicated in MMTV infection during oral transmission using mouse models that have significantly reduced levels of M cells (*Golovkina et al., 1999*). Thus, we asked whether incoming MLV infiltrates PP through M cells. We infected 3-day-old mice with MLV Env-Nluc, sampled intestines at 48 hpi to identify luciferase-positive PPs, and processed them for electron tomography to delineate possible infiltration mechanisms (*Figure 7A*). MLV particles were observed within endosomes inside of M cells (*Figure 7B*). These data supported a contribution of M cells and a transcytosis model for retrovirus infiltration from the intestinal lumen into the follicle region of the PP (*Bomsel and Alfsen, 2003*; *Kobayashi et al., 2019*; *Bomsel, 1997*).

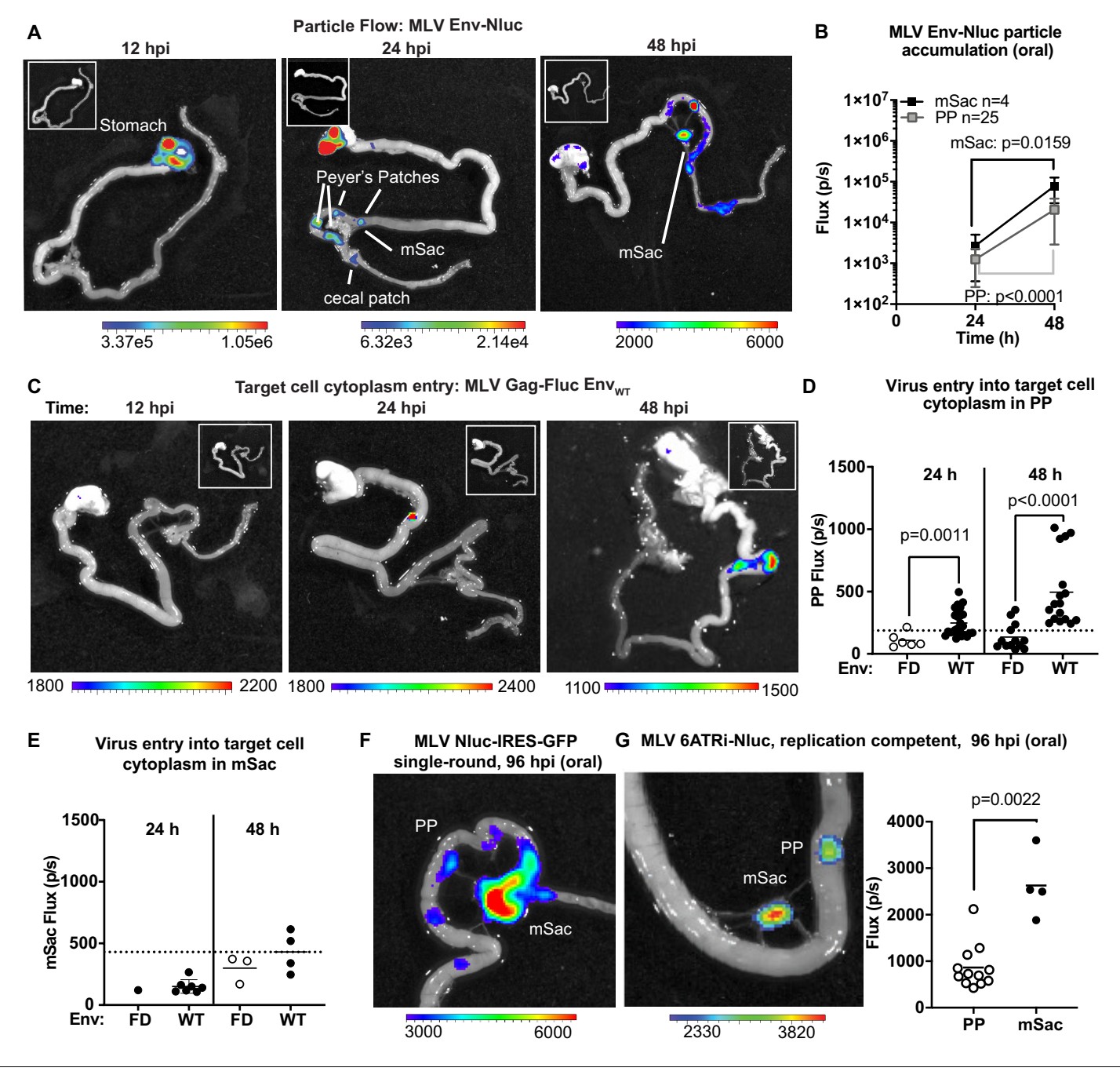

**Figure 6.** Real-time visualization of individual steps of retrovirus infection in vivo during oral challenge. (**A**) Overlaid bioluminescence images of gastrointestinal tract from neonatal B6 mice orally challenged with $1 \times 10^6$ IU of replication-defective, luciferase-labeled Env-Nluc murine leukemia virus (MLV) particles at 3 days of age. Mice were sacrificed at the indicated time points and their gastrointestinal tracts were subjected to bioluminescence imaging (BLI) for monitoring the sequential flow of incoming virus particles through stomach, and developing Peyer's patches (PP), mesenteric sac (mSac), and cecal patch. (**B**) Temporal accumulation of orally challenged MLV Env-Nluc displayed as Flux (photons/s) for an experiment as in (**A**), in PP (n = 25) and mSacs (n = 4). Error bars denote standard deviation. (**C**) Overlaid bioluminescence images of gastrointestinal tract from neonatal B6 mice orally challenged with $2 \times 10^4$ relative light units (RLU) of MLV Gag-Fluc $Env_{WT}$ at 3 days of age. Mice were sacrificed at the indicated time points, and their gastrointestinal tracts were subjected to BLI for monitoring virus entry and access to target cell cytoplasm. (**D, E**) Quantification of MLV cytoplasmic entry at indicated times in PP (**D**; n = 6–19) and mSacs (**F**; n = 1–8) displayed as Flux (photons/s) from the experiment described in (**C**). Neonatal B6 mice orally challenged with $2 \times 10^4$ RLU of MLV Gag-Fluc $Env_{FD}$ were used as control. Dotted lines indicate three standard deviations above background (uninfected mice) means at each location. (**F, G**) Overlaid bioluminescence images of gastrointestinal tract from neonatal B6 mice orally challenged with $1 \times 10^6$ IU of single-round MLV pMIG-Nluc-IRES-GFP (**G**) or full-length replication-competent MLV 6ATRi-Nluc at 3 days of age.

*Figure 6 continued on next page*

*Figure 6 continued*
Mice were sacrificed at 96 hr post infection, and their gastrointestinal tracts were subjected to BLI for visualizing infection in PP and mSacs. The plot shows quantification of signal at PP and mSac for an experiment as in accompanying image. p values were derived from non-parametric Mann–Whitney test; mean values are denoted by horizontal line. Scale bars shown for BLI denote radiance in photons per second per square centimeter per steradian ($p/s/cm^2/sr$).
The online version of this article includes the following figure supplement(s) for figure 6:

**Figure supplement 1.** Characterization of infected cells in the mesenteric sacs (mSacs) of neonatal mice.

## CD169 contributes to virus particle accumulation and establishment of infection in the GI tract

We next asked if viruses are captured after arriving in the PP. Visualizing virus particles (MLV Gag-GFP) at the single-cell level by immunostaining of tissue sections was challenging after particles had traveled for days in the GI tract before arriving in the PP follicle. We therefore resorted to performing retrovirus challenge in adult mice by surgically ligating a region in the small intestine that contained PP. The ligated loop allowed us to increase the local particle concentration of MLV Gag-GFP under live settings (*Jia and Edelblum, 2019*). After allowing the virus to be taken up for 1 hr, PP region of the ligated intestine was processed for immunostaining and microscopy (*Figure 8*). In addition to particles that were free or adhered to the epithelial cells, we observed particles invading the epithelial barrier presumably through M cells and dispersed in the PP follicle (*Figure 8A*). CD169[+] macrophages were not previously reported in the PP. Surprisingly, we observed CD169[+] macrophages located in the serosal side of the PP that had captured MLV Gag-GFP (*Figure 8B*). These

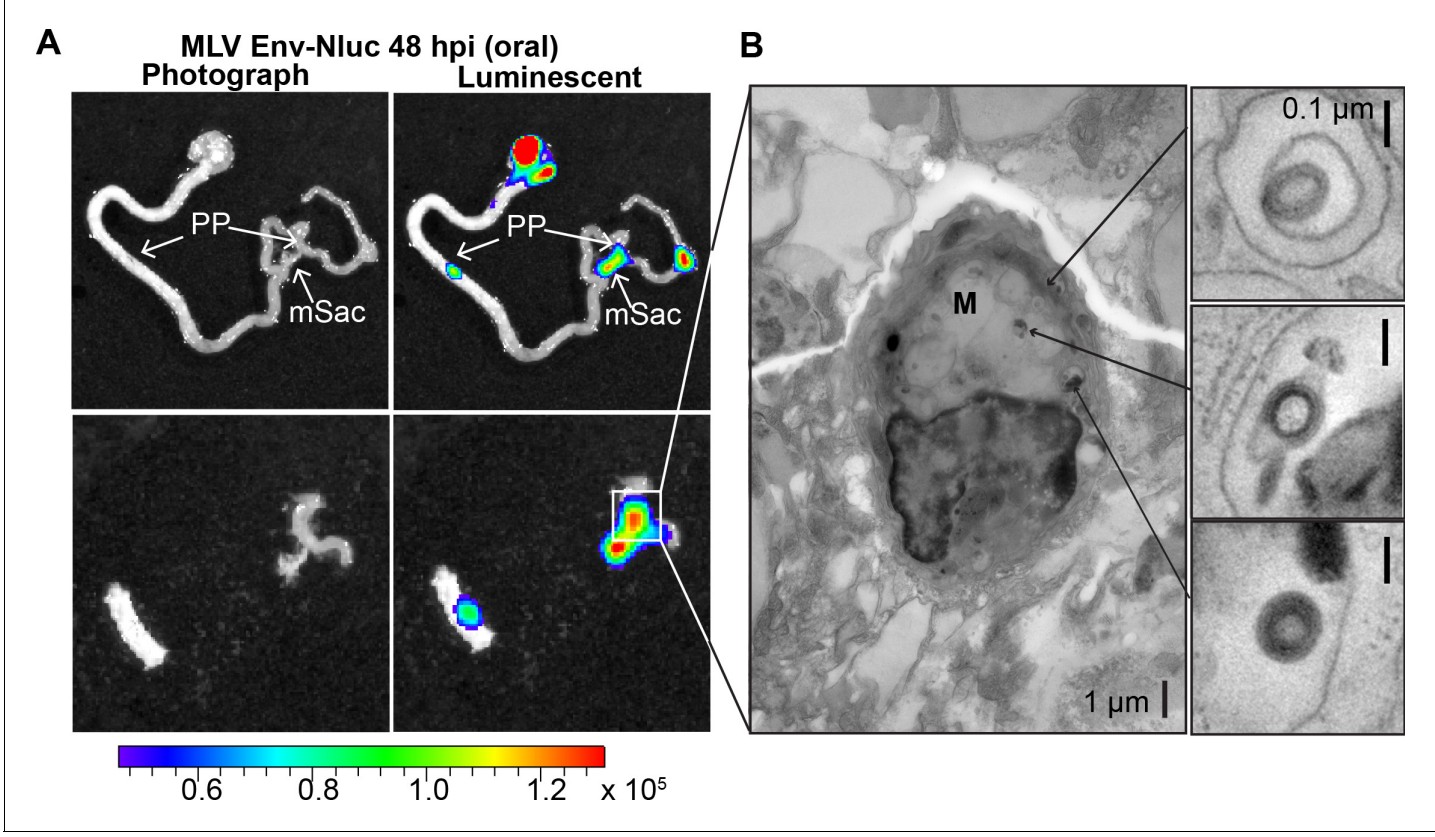

**Figure 7.** Murine leukemia virus (MLV) infiltrates intestinal Peyer's patches (PP) and can be found in endosomes of microfold (M) cells after oral challenge. (**A**) Bioluminescence imaging (BLI) of gastrointestinal tract isolated from a neonatal B6 mice 48 hr after oral challenge with $1 \times 10^6$ IU of MLV Env-Nluc at 3 days of age, showing luciferase-positive PP and mesenteric sac (mSac). Scale bars shown for BLI denote radiance in photons per second per square centimeter per steradian ($p/s/cm^2/sr$) (**B**) EM tomogram and magnified insets of BLI-identified PP as in (**A**), showing MLV within endosomes of M cells. Scale bars as indicated.

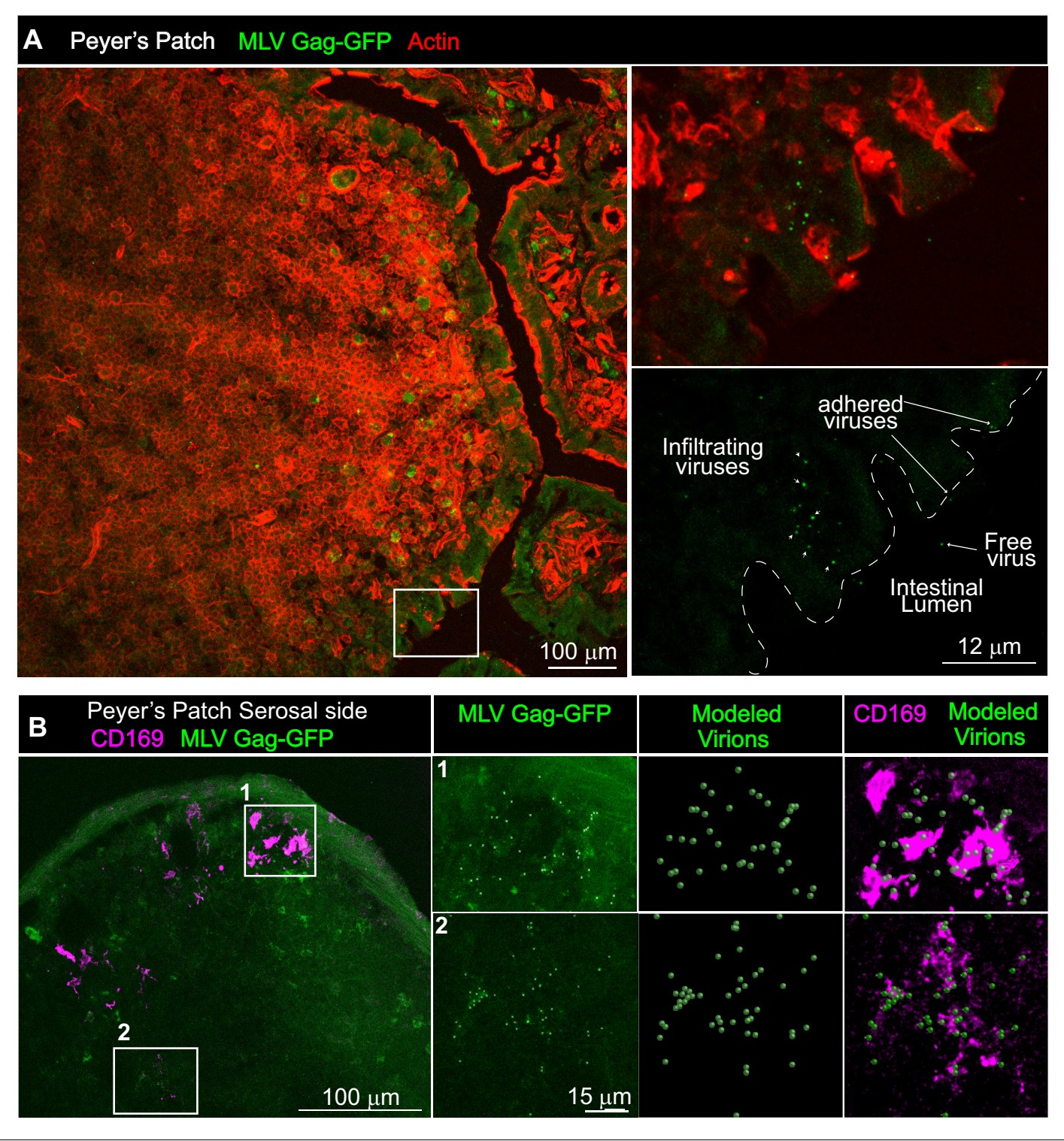

**Figure 8.** Gut-infiltrating murine leukemia virus (MLV) is captured by Peyer's patch (PP)-resident CD169[+] macrophages. (A) Images of cryosections from ligated gut tissue containing a PP from B6 mice that was challenged for 1 hr with MLV Gag-GFP (green) corresponding to $1$–$4 \times 10^5$ IU based on comparative western blot analyses with antibodies to Gag with equivalent amounts of WT FrMLV. Tissue sections were stained with phalloidin (red) to visualize actin in the PP tissue structure. Dotted lines demarcate the epithelium from intestinal lumen. Free, adhered, and infiltrating MLV Gag-GFP particles are as indicated with arrows. (B) Images of cryosections from PP for an experiment as in (A). Macrophages identified using antibodies to CD169 (magenta) were predominantly located at the serosal side of PP. MLV Gag-GFP (green) viruses within marked insets of PP were identified and modeled into spheres (modeled virions), which were then merged to depict close association with CD169[+] macrophages. Scale bars as indicated.

data suggested a possible role for CD169[+] macrophages in capturing and promoting retrovirus infection in intestinal PPs. We explored this possibility further in neonatal mice. In contrast to their serosal location in adult PPs, CD169[+] macrophages were more dispersed within developing neonatal PPs (*Figure 9A*). We next asked whether CD169 plays a functional role in promoting retrovirus acquisition during oral transmission from mother to offspring. We first orally inoculated B6 and CD169[-/-] mice with MLV Env-Nluc to monitor virus particle transit through the intestine. BLI at 48 hpi revealed a significantly reduced distribution of virus particles throughout the intestines and within PP (approximately fivefold reduction; p<0.0001) as well as mSacs (approximately threefold reduction; p=0.0082) in CD169[-/-] mice compared to B6 mice (*Figure 9B–D*). We then asked whether CD169 promoted infection in the mSac. We infected neonatal B6 and CD169[-/-] mice orally with WT FrMLV and assessed infection levels in individual mSacs via flow cytometry for viral GlycoGag protein expression at 5 dpi. Our analyses showed that CD169[-/-] mice displayed a 26-fold reduction in the establishment of infection in the mSac compared to B6 mice (p=0.0011) (*Figure 9E*). These results revealed that CD169 is a host factor that contributes to retroviral acquisition via the oral transmission route.

## Discussion

We developed a whole-body imaging-guided, top-down approach to study retrovirus infection in vivo. We engineered a series of reporter viruses to monitor individual steps of virus infection including virus particle flow, entry into cytoplasm, infection of cells, replication, and spread in the context of living animals. We first validated this system by successfully demonstrating its ability to uncover new insights into well-studied subcutaneous and intravenous transmission routes. Importantly, BLI permitted temporal tracking of orally administered retroviruses along the GI tract as they traversed the lumen through PP to reach the draining mSac. We have also shown that a common host factor expressed on sentinel macrophages, CD169, promotes retrovirus capture and acquisition throughout three routes. Our results highlighted how retroviruses co-opt the immune surveillance function of tissue-resident sentinel macrophages to establish infection. Our visual approach to virus infection in vivo revealed several novel facets of host-virus interplay that we discuss below.

### Route-specific tissue barriers

Each route of entry comprises a unique set of tissue-specific barriers that viruses must overcome for successful infection of the host. Our particle flow analyses for the retroorbital route revealed that most viruses reach the spleen, where they establish infection after briefly passing through the heart and liver. In contrast, the i.f.p. route presented a much more challenging barrier for viruses to reach their target organ, the pLN. Virus accumulation in the footpad was 150–200-fold higher than that in the pLN (*Figure 3B*). We frequently observed clusters of incoming virus particles trapped within the dense, tightly packed matrix of footpad muscle and collagen fibrils. In addition, virus particles were taken up by macrophages and DCs that presumably trap them in a non-productive pathway in contrast to B and T cells that when infected serve as amplifying hosts for MLV. In addition, virus-associated sentinel macrophages and DCs are known to collaborate and initiate immune responses, as has been shown for other viruses (Influenza, VSV) and viral antigens during i.f.p. challenge (*Iannacone et al., 2010*; *Junt et al., 2007*; *Chatziandreou et al., 2017*; *Gonzalez et al., 2010*; *Heesters et al., 2013*; *Moseman et al., 2012*). Despite the high levels of particle capture and infection seen near the injection site at the footpad, our previous studies showed that the establishment of infection at the draining pLN is independent of cells migrating from the footpad (*Uchil et al., 2019b*). During oral transmission, enveloped retroviruses must withstand harsh conditions such as low pH in the stomach, digestive enzymes, and bile salts (*Pfeiffer, 2010*; *Pirtle and Beran, 1991*) that can solubilize viral membranes. While retroviruses can be destroyed at pH as high as 4 (*Ye et al., 2003*), MLV is stable until pH 3 (*Wallin et al., 2004*). Furthermore, mice are most susceptible to MLV infection at day 3 post-partum (*Buffett et al., 1969b*) when stomach acid production is low (*Deren, 1971*). Moreover, milk may also shield retroviruses from acid secreted in the neonatal stomach. Our analyses comparing the flux of input MLV Env-Nluc viruses (~1 × 10[7] p/s) with that of signal seen at 48 hr in PP and mSac (~1–2 × 10[5] p/s) (*Figure 6B*) suggest that ~50–100-fold fewer viruses are able to surmount the oral and GI barriers. This barrier is expected to increase significantly

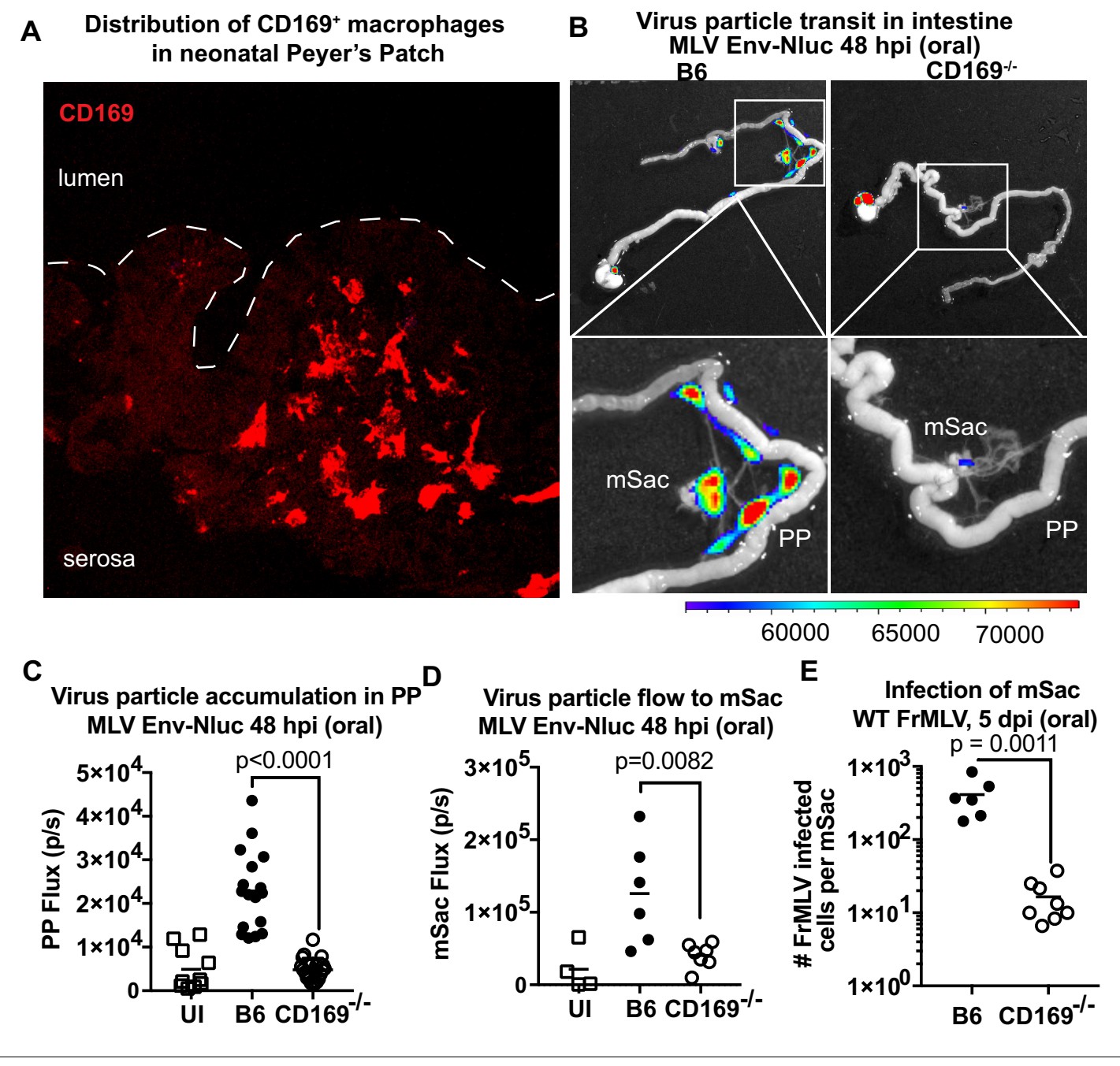

**Figure 9.** CD169 contributes to retrovirus particle acquisition and establishment of infection during oral challenge. (**A**) Image of a Peyer's patch (PP) cryosection from a 3-day-old neonatal B6 mouse. Macrophages in the developing follicle were identified using antibodies to surface marker CD169 (red), and dotted lines demarcate the epithelium from intestinal lumen. (**B**) Representative images of gastrointestinal tracts isolated from a neonatal B6 and CD169$^{-/-}$ mice 48 hr after oral challenge with $1 \times 10^6$ IU of murine leukemia virus (MLV) Env-Nluc at 3 days of age to show comparative accumulation of bioluminescent viruses in PP and mesenteric sac (mSac). Scale bars shown for bioluminescence imaging denote radiance in photons per second per square centimeter per steradian (p/s/cm$^2$/sr). (**C, D**) Quantification of virus transit to PP (**C**) and mSacs (**D**) at 48 hr post infection (hpi) in B6 and CD169$^{-/-}$ mice from the experiment described in (**B**). Virus transit was quantified as Nluc photon flux (photons/s). (**E**) FrMLV-infected cells 5 dpi (oral., $1 \times 10^6$ IU) in mSac (n = 6–8) from neonatal B6 and CD169$^{-/-}$ mice challenged at 3 days of age. Single-cell suspensions of cells from individual mSacs were obtained at 5 dpi and processed for flow cytometry. Infection levels were determined using antibodies to FrMLV GlycoGag. p values derived from non-parametric Mann–Whitney test; mean values denoted by horizontal line.

with age due to the upsurge in production of virus-inactivating factors like bile salts and acid, thus rendering adult mice resistant to oral MLV transmission (*Deren, 1971*).

## Differential kinetics of virus particle arrival and cytoplasmic entry

Following i.v. infection, viruses accumulated in the spleen within 3 min and entered the cytoplasm of host target cells by 60 min post-challenge (*Figure 2D, E*). In footpads, cytoplasmic entry corresponded with virus particle arrival (*Figure 3A–D*). In pLNs, however, viruses took longer (more than 40 min) to enter the host cell cytoplasm after arrival, with pLN luminescence first observed at 6 hr post-challenge (*Figure 3C, E*). Similarly, we saw a weak but delayed cytoplasmic entry of viruses in the mSac compared to PP despite near simultaneous arrival of virus in both the organs (*Figure 6B, D, E*). The delayed cytoplasmic entry in the pLN and mSacs is likely explained by the mode of infection. We have previously documented that in the spleen and pLN, CD169[+] macrophages capture incoming viruses and promote infection of target lymphocytes by a process called *trans*-infection (*Sewald et al., 2015*; *Uchil et al., 2019b*; *Pi et al., 2019*). Sentinel macrophages are resistant to MLV infection at early time points as viruses are held at a distance of ~41 nm from the cell surface, corresponding to the length of the CD169 ectodomain, a distance too far for the MLV Env to engage the receptor (*Sewald et al., 2015*). Following capture, viruses must transit through surface-associated membrane invaginations of CD169[+] sentinel macrophages before presentation to target cells for *trans*-infection, delaying cytoplasmic entry. In contrast, in footpads, incoming virus may directly interact with DCs in addition to the hand-off from virus-capturing macrophages, as observed by our immunohistochemistry and electron tomography studies (*Figure 4B, C*). Consistent with our observations of rapid virus entry at the footpad, virus-capturing DCs may internalize virus and become infected to perform their natural role of antigen presentation unlike CD169[+] macrophages, which cross-present viruses (*Uchil et al., 2019b*; *van Dinther et al., 2018*). Indeed, a large proportion of infected cells in the footpads were CD11b[+]CD11c[+] DCs. This likewise may occur within PPs, where viruses, after transiting through M cells, have direct access to target cells in the underlying follicle in addition to availing CD169[+] macrophage-promoted infection. In contrast, free viruses entering mesenteric LN undergo sentinel macrophage-mediated *trans*-infection, delaying cytoplasmic access like those seen in the pLN.

## Differential capture of retroviruses by CD169 in spleen and liver

We observed that retroorbitally administered viruses briefly passed through the liver before accumulating in the spleen (*Figure 2A, B*). The vastly reduced virus retention in the liver, despite the presence of sinusoidal CD169[+] Kupffer cells, was surprising. However, these results could be explained by several fold lower levels of CD169 expression in Kupffer cells compared to CD169[+] SCS macrophages in the LN or marginal zone metallophilic macrophages in the spleen (*Saunderson et al., 2014*). A similar predominant capture of intravenously delivered exosomes in spleen compared to liver was also observed previously (*Saunderson et al., 2014*). Alternatively, specific tissue environments may govern the capacity of lectins to bind incoming viruses as the binding capacity of Siglecs is often regulated by endogenous ligands (*Varki and Gagneux, 2012*; *Crocker and Varki, 2001*).

### Novel facets of oral retroviral transmission

While requiring necropsy to increase resolution and sensitivity, BLI assisted the study of long organs such as the GI tract as it can illuminate areas of interest to guide directed investigations and shed light on the sequential course of infection events. Particle flow analyses with replication-defective reporter viruses demonstrated that it takes 24–48 hr for viruses to reach portals of entry (PP) and accumulate in the draining mSac. Interestingly, prior replication in PP was not required for transiting to the draining LN (*Figure 6A–C*). Incoming particles were able to access the lymph flow for transit to the mSac. This contrasts with vaginal infection of SIV, in which local replication was critical for further virus dissemination (*Haase, 2010*; *Haase, 2011*; *Li et al., 2009*). The use of Fluc-tagged viruses revealed that entry into the cytoplasm occurs at ~24 hpi in the PP and begins at ~48 hpi in the mSac, which houses mesenteric LNs (*Figure 6D*). This was followed by infection at 96 hpi (*Figure 6F, G*). Interestingly, in addition to B and CD4[+] T cells that together constituted 40–50% of infected cells in neonatal mSacs, CD11c[+] DCs were the predominant infected cell population (~50%). DCs are known to migrate from periphery to draining LN after antigenic stimulus. It is likely that most of the infected

DCs originated from PP and migrated to mSacs, as has been proposed for other retrovirus infections (*Dudley et al., 2016*; *Beutner et al., 1994*; *Held et al., 1993*). As with other infection routes we tested, CD169 expression on sentinel macrophages was crucial for promoting oral transmission (*Figure 9*). Particle retention and subsequent infection at PP and mSacs were significantly reduced in the absence of CD169. These data were reminiscent of our previous study, where we observed a CD169 requirement to promote MLV infection at both pLN and spleen (*Sewald et al., 2015*; *Uchil et al., 2019b*). Thus, our studies revealed the existence of CD169-mediated capture and infection promotion as a second crucial step downstream of likely entry through M cells that augments oral retroviral transmission.

Overall, our BLI-guided analyses have highlighted how retroviruses, during a million years of co-evolution, have co-opted CD169, primarily used for immune surveillance by tissue-resident sentinel macrophages, as a common host factor for promoting host colonization via various naturally occurring infection routes. Our study opens avenues for a localized CD169-blockade-based strategy to curb retrovirus acquisition and transmission.

# Materials and methods

## Key resources table

| Reagent type (species) or resource | Designation | Source or reference | Identifiers | Additional information |
|---|---|---|---|---|
| Antibody | Fc block anti mouse-CD16/CD32 (Rat monoclonal) | BioLegend | Cat # 101302 RRID:AB_312801 | FACS (1:100) |
| Antibody | Anti-MLV Glycogag (mab34) (Rat monoclonal) | Santiago Lab/ Bruce Chesebro | Recognizes MLV GlycoGag | (1:2000) |
| Antibody | AF647 or A488 anti-MLV GlycoGag (mab34) (Rat monoclonal) | Purified from hybridoma | Antibody labeled using commercial kit | FACS (1:2000) IF (1:300 µl) |
| Antibody | PE/Cy7 anti-mouse CD19(6D5) (Rat monoclonal) | BioLegend | Cat # 115507 RRID:AB_313654 | FACS (1:1000) |
| Antibody | PE/Cy7 anti-mouse CD4 (GK1.5) (Rat monoclonal) | BioLegend | Cat # 100421 RRID:AB_312706 | FACS (1:1000) |
| Antibody | APC anti-mouse CD3(145–2 C11) (Hamster monoclonal) | BioLegend | Cat # 100312 RRID:AB_312677 | FACS (1:500) |
| Antibody | PE anti-mouse/human CD11b (M1/70) (Rat monoclonal) | BioLegend | Cat # 101208 RRID:AB_312791 | FACS (1:500) |
| Antibody | AF647 anti-mouse CD169 (3D6.112) (Rat monoclonal) | BioLegend | Cat # 142407 RRID:AB_2563620 | FACS (1:500) |
| Antibody | AF594 anti-mouse CD169 (3D6.112) (Rat monoclonal) | BioLegend | Cat # 142416 RRID:AB_2565620 | IF (1 in 300 µl) |
| Antibody | AF647 anti-mouse CD11c (N418) (Hamster monoclonal) | BioLegend | Cat # 117314 RRID:AB_492850 | FACS (1:500) |
| Antibody | APC Rat anti-mouse CD45 (30-F11) (Rat monoclonal) | BD-Pharmingen | Cat # 559864 RRID:AB_398672 | FACS (1:500) |
| Antibody | Alexa Fluor 594 anti-mouse CD11c Antibody (Hamster monoclonal) | BioLegend | Cat # 117346 RRID:AB_2563323 | IF (1 in 300 µl) |

*Continued on next page*

*Continued*

| Reagent type (species) or resource | Designation | Source or reference | Identifiers | Additional information |
|---|---|---|---|---|
| Antibody | APC-Cy7 anti-mouse CD11c (N418) (Hamster monoclonal) | BioLegend | Cat #117324 RRID:AB_830649 | FACS (1:500) |
| Chemical compound, drug | Liberase TL Research Grade | Sigma-Aldrich | Cat# 5401020001 | 0.2 mg/ml |
| Chemical compound, drug | DNAse I recombinant, RNAse-free | Roche | Ref # 04716728001 | 20 µg/ml |
| Chemical compound, drug | RBC Lysis Buffer (10X) | BioLegend | Cat # 420301 | |
| Chemical compound, drug | Bovine serum albumin (BSA) | Sigma-Aldrich | Cat# A9647-100G CAS: 9048-46-8 | |
| Chemical compound, drug | Accutase | BioLegend | Cat # 423201 | |
| Chemical compound, drug | Gelatin (Teleostean gelatin) Type A | Sigma-Aldrich | Cat # G7041 CAS: 9000-70-8 | |
| Chemical compound, drug | Triton-X 100 t-octyl phenoxy polyethoxyethanol | American Bioanalytical | Cat # AB02025-00500 CAS: 9002-93-1 | |
| Chemical compound, drug | L-lysine Monohydrochloride | Sigma-Aldrich | Cat # L1262 | |
| Chemical compound, drug | Sodium (meta)periodate | Sigma-Aldrich | Cat # 30323–100G CAS: 7790-28-5 | |
| Chemical compound, drug | Tissue-Tek O.C.T Compound | Sakura | Cat # 4583 | |
| Chemical compound, drug | Fc receptor blocker | Innovex | Cat # NB335-5 | |
| Chemical compound, drug | ProLong Gold antifade reagent | Invitrogen | Cat # P36934 | |
| Chemical compound, drug | Glutaraldehyde | Electron Microscopy Sciences | Cat # 16220 CAS: 111-30-8 | |
| Chemical compound, drug | Sodium cacodylate trihydrate | Electron Microscopy Sciences | Cat #12300 | |
| Chemical compound, drug | Ficoll | Sigma-Aldrich | Cat #F2878-100g | |
| Chemical compound, drug | Osmium tetroxide | Electron Microscopy Sciences | Cat #19110 | |
| Chemical compound, drug | Uranyl acetate | Electron Microscopy Sciences | Cat #22400 | |

*Continued on next page*

Continued

| Reagent type (species) or resource | Designation | Source or reference | Identifiers | Additional information |
|---|---|---|---|---|
| Chemical compound, drug | Acetone, EM-Grade, Glass-Distilled | Electron Microscopy Sciences | Cat #10015 | |
| Chemical compound, drug | Epon-Araldite resin | Electron Microscopy Sciences | Cat #13940 | |
| Chemical compound, drug | Lead citrate | Electron Microscopy Sciences | Cat #17800 CAS: 512-26-5 | |
| Chemical compound, drug | Gold beads (10 nm) | Ted Pella, Inc | Cat. #15703-1 | |
| Chemical compound, drug | Dimethyl sulfoxide (DMSO) | Sigma-Aldrich | Cat # D2650-5X5ML CAS: 67-68-5 | |
| Chemical compound, drug | Sodium azide | Sigma-Aldrich | Cat # S-8032 EC No: 247-852-1 | |
| Chemical compound, drug | Passive lysis buffer (5X) | Promega | Cat # E194A | |
| Chemical compound, drug | D-Luciferin, Potassium Salt (Proven and Published) | Gold Biotechnology | Cat #LUCK-3G | 15 mg/ml solution 5 µl/g body weight |
| Chemical compound, drug | Hoechst 33342 | Invitrogen | Cat # H3570 | |
| Commercial assay or kit | Mix-n-Stain CF 488A Antibody Labeling Kit (50–100 µg) | Sigma-Aldrich | Cat # MX488AS100 SIGMA | |
| Commercial assay or kit | Mix-n-Stain CF 647 Antibody Labeling Kit (50–100 µg) | Sigma-Aldrich | Cat # MX647S100 SIGMA | |
| Commercial assay or kit | Nano-Glo Luciferase Assay System | Promega | Cat # N1120 | Diluted (1:40) in 1X PBS, 5 µl/g body weight |
| Commercial assay or kit | KAPA SYBR FAST qPCR Master Mix (2X) Kit | KAPA Biosystems | Cat # KK4600 and KK4601 | |
| Commercial assay or kit | RNeasy Mini Kit (50) | Qiagen | Cat #/ID 74104 | |
| Commercial assay or kit | Gibson Assembly Kit | NEB | Cat #E5520S | |
| Commercial assay or kit | iQ Multiplex Powermix | Bio-Rad | Cat # 1725848 | |
| Commercial assay or kit | iScript cDNA Synthesis Kit | Bio-Rad | Cat # 95047-100 | |
| Cell line (*Homo sapiens*) | HEK293 | ATCC | Cat # CRL-1573 RRID:CVCL_0045 | |
| Cell line (*Gallus gallus domesticus*) | DFJ8 | Mothes Lab | | From Jim Cunningham, Dana Farber; target cells for determining MLV titer |
| Sequence-based reagent | Nluc_F | This work, *Figure 2* | PCR primers | GGAGGTGTGTCC AGTTTGTT |

*Continued on next page*

*Continued*

| Reagent type (species) or resource | Designation | Source or reference | Identifiers | Additional information |
|---|---|---|---|---|
| Sequence-based reagent | Nluc_R | This work, *Figure 2* | PCR primers | ATGTCGATCTTCAG CCCATTT |
| Sequence-based reagent | FrMLV Gag_ F | This work, *Figure 2* | PCR primers | GAGAGAGGGGAG GTTTAGGGT |
| Sequence-based reagent | FrMLV Gag_ R | This work, *Figure 2* | PCR primers | AAGGCGCTGGG TTACATTCT |
| Recombinant DNA reagent | pLRB303-FrMLV-6ATRI-Nluc | This work, *Figures 1*, *2*, *3* , *6* and *9* | | Mothes Lab, used for monitoring virus replication and dissemination; see Materials and methods |
| Recombinant DNA reagent | pLRB303-FrMLV-Env-PRR-Nluc | This work, *Figure 1* | | Mothes Lab, used for monitoring virus particle flow; see Materials and methods |
| Recombinant DNA reagent | pLRB303-FrMLV-Gag-Fluc-Env$_{wt}$ | This work, *Figures 1*, *2*, *3* and *6* | | Mothes Lab, used for monitoring virus fusion; see Materials and methods |
| Recombinant DNA reagent | pLRB303-FrMLV-Gag-Fluc-Env$_{FD}$ (SFFV gp55) | This work, *Figures 1*, *2*, *3* and *6* | | Mothes Lab, control for monitoring virus fusion; see Materials and methods |
| Recombinant DNA reagent | pLRB303-FrMLV-Gag-GFP | Mothes Lab *Sewald et al., 2015* | | Mothes Lab, for labeling and visualizing MLV virions |
| Recombinant DNA reagent | pcDNA3.1 FrMLV Env PRR-Nluc | This work *Figures 1*, *2*, *3*, *6*, *7* and *9* | | Mothes Lab, used for monitoring virus particle flow; see Materials and methods |
| Recombinant DNA reagent | pMIG-Antares | This work, *Figure 2* | | Mothes Lab, for co-packaging Antares reporter in MLV; see Materials and methods |
| Recombinant DNA reagent | pGL4.32[luc2P NF-kB-RE] | Promega, Madison, WI | Cat # E8491 | Template to amplify Fluc |
| Recombinant DNA reagent | pNL1.1 | Promega, Madison, WI | RRID:Addgene_141285 | Template to amplify Nluc |
| Recombinant DNA reagent | pMMP-LTR-GFP | Mothes Lab | | From Jim Cunningham, Dana Farber; for co-packaging GFP reporter in MLV |
| Recombinant DNA reagent | pMIG-Nluc-IRES-GFP | Mothes Lab *Ventura et al., 2019* | | |
| Recombinant DNA reagent | pMIG-Fluc-IRES-mCherry | Addgene | RRID:Addgene_75020 | |
| Recombinant DNA reagent | pNCS-Antares | Addgene | RRID:Addgene_74279 | |
| Recombinant DNA reagent | pMIG-w | Addgene | RRID:Addgene_12282 | |
| Software, algorithm | Accuri CSampler | BD Biosciences | RRID:SCR_014422 | Analyses of flow cytometric data |
| Software, algorithm | FlowJo | Treestar | RRID:SCR_008520 | Analyses of flow cytometric data |
| Software, algorithm | Nikon-Elements AR Analysis v4.13 and Acquisition v4.5 | Nikon | | Image analyses |

*Continued on next page*

*Continued*

| Reagent type (species) or resource | Designation | Source or reference | Identifiers | Additional information |
|---|---|---|---|---|
| Software, algorithm | CFX Maestro | Bio-Rad Inc | RRID:SCR_017251 | qPCR analyses |
| Software, algorithm | Living Image v4.7.3 | PerkinElmer | RRID:SCR_014247 | Analyzes software for bioluminescence imaging |
| Software, algorithm | GraphPad Prism v9.0.1 | GraphPad Software | RRID:SCR_002798 | https://www.graphpad.com/ |
| Software, algorithm | IMOD | David N. Mastronarde, University of Colorado Boulder | RRID:SCR_003297 | https://bio3d.colorado.edu/imod/ |
| Software, algorithm | SerialEM | David N. Mastronarde, University of Colorado Boulder | RRID:SCR_017293 | https://bio3d.colorado.edu/SerialEM/ |
| Other | TriStar LB 941 Multimode Microplate Reader and Luminometer | Berthold Technologies GmbH and Co. KG | | |
| Other | BD Biosciences C6 Accuri Flow Cytometer | BD Biosciences | RRID:SCR_019591 | Flow cytometer |
| Other | PerkinElmer IVIS Spectrum In-Vivo Imaging System | PerkinElmer | RRID:SCR_018621 | Yale University ABSL-3 and ABSL-2 facility |
| Other | XGI-8 Gas Anesthesia System | Perki Elmer | | Yale University ABSL-3 and ABSL-2 facility |
| Other | Leica Cryostat CM1950 | Leica | RRID:SCR_018061 | CM1950 (Akiko Iwasaki Lab) |
| Other | Nikon CSU-W1 Spinning Disk Confocal microscope | Nikon Instruments Inc, Americas | | Yale West Campus Imaging Core |
| Other | Leica TCS DMi8 SP8 microscope | Leica | | CCMI Yale Central Facility |
| Other | Transmission electron microscope | Tecnai | TF30ST-FEG | |
| Other | C1000 Touch thermal cycler | Bio-Rad | Cat # 1851148 | PCR machine |
| Other | CFX Connect Real-Time PCR Detection System | Bio-Rad | Cat # 1855201 | Real-time PCR machine |

## Mice

C57BL/6J (B6) (RRID:IMSR_JAX:000664) and BALB/cJ (RRID:IMSR_JAX:000651) mice were obtained from the Jackson Laboratory (Bar Harbor, ME). CD169$^{-/-}$ mice (B6 background; RRID:IMSR_EM:12765) were from Paul Crocker (*Oetke et al., 2006*), University of Dundee, UK. Animals were housed under specific pathogen-free conditions in the Yale Animal Resources Center (YARC) in the same room of the vivarium. The Institutional Animal Care and Use Committees (IACUC) and the Institutional Biosafety Committee of Yale University approved all experiments. 6–8-week-old male and female mice were used for all experiments involving adult mice. Breeder mice acting as foster mothers for litter transfer experiments were 3–6 months of age. Oral inoculation experiments in neonatal mice were performed on 3-day-old mice.

## Generation of viral vector plasmids

Virus-encoding plasmids were generated using Gibson Assembly (NEB Gibson Assembly Kit, NEB, Ipswich, MA). Insert amplicons containing 25 bp overlaps to target regions were generated using Kapa HiFi Hotstart high-fidelity polymerase (Kapa Biosystems/Sigma-Aldrich, St. Louis, MO) using a

touchdown PCR protocol. MLV Env-Nluc: For generating full-length MLV Env-Nluc construct, MLV pLRB303 Env-PRR-GFP plasmid encoding a full-length MLV genome (*Lehmann et al., 2005*) was digested with Nhe I to release the GFP cloned into the PRR of MLV envelope. Nluc was amplified from the pNL1.1 plasmid (Promega, Madison, WI) with primers to ensure in-frame Nluc expression as well as containing 25 bp overlap to the insertion site to allow cloning by Gibson Assembly. A similar strategy was used to generate MLV envelope expressor plasmid for insertion of Nluc in-frame into PPR region. MLV Gag-GFP: MLV Gag-GFP in the full-length viral context was described previously (*Sewald et al., 2015*). MLV Gag-Fluc Env$_{WT}$: GFP was released from MLV Gag-GFP using EcoRI and HindIII digestion. Fluc was amplified from pGL4.32[luc2P NF-kB-RE] plasmid (Promega) and inserted into the vector backbone using Gibson Assembly. For MLV Gag-Fluc Env$_{FD}$ generation, MLV envelope was replaced with SFFV envelope using restriction digestion and ligation. SFFV envelope was amplified from pBR322-SFFV LS, a gift from Leonard Evans and Frank Malik (NIH). Full-length MLV 6ATRi-Nluc:gBlocks encoding 6ATRi-Nluc along with flanking region were obtained from Integrated DNA Technologies (Coralville, IA). Amplified gBlocks and full-length Friend MLV plasmid were digested with ClaI and BlpI and assembled into the FrMLV backbone via Gibson Assembly. pMIG-Nluc-IRES-GFP: pMIG-Nluc-IRES-GFP was described previously (*Ventura et al., 2019*). pMIG-Nluc-IRES-GFP  was described previously (*Ventura et al., 2019*). pMIG-Fluc-IRES-mCherry was a gift from Xiaoping Sun (Addgene plasmid # 75020; http://n2t.net/addgene:75020; RRID:Addgene_75020). pMIG-Antares: pMIG-Antares was generated by replacing the IRES GFP cassette of pMIG-w with Antares luciferase from the pNCS-Antares. pMIG-w was a gift from Luk Parijs (Addgene plasmid # 12282; http://n2t.net/addgene:12282; RRID:Addgene_12282) (*Li et al., 2009*), and pNCS-Antares was a gift from Michael Lin (Addgene plasmid # 74279; http://n2t.net/addgene:74279; RRID:Addgene_74279) (*Chu et al., 2016*). Plasmids were transformed into DH5α Max Efficiency competent *Escherichia coli* (Thermo Fisher, Waltham, MA). *E. coli* were grown overnight in 1 l cultures of 2X Yeast extract Tryptone (YT) media at 30–37°C under shaking conditions. Plasmids were isolated using Machery-Nagel DNA preparation kits. Reporter gene expression was tested by transfecting 50 ng of plasmid and 450 ng of pcDNA3.1 into HEK293 cells (ATCC Cat# PTA-4488, RRID:CVCL_0045) seeded in 24-well plates. 24 hr post-transfection, reporter gene expression was monitored via flow cytometry and/or luciferase assay.

## Generation of viral stocks

Plasmids encoding each vector were transfected into HEK293 cells (ATCC; checked routinely for *Mycoplasma* contamination using MycoSensor Assay Kits, Agilent Inc) seeded in 10 cm plates using Fugene 6 (Promega) or polyethyleneimine. 12 μg of plasmid encoding full-length viral vectors were transfected into each 10 cm plate. Replication-defective viruses were produced via transfection of 12 μg total DNA consisting of a mixture of MLV Gag-Pol, LTR-Reporter plasmid, and Ecotropic MLV Envelope in a 5:5:2 ratio. Replication- Gag-Fluc Env$_{WT}$ viruses were generated as above but with 8 μg of full-length FrMLV plasmid encoding Gag-Fluc, 2 μg of MLV Gag-Pol, 2 μg LTR-GFP plasmids. Gag-Fluc Env$_{FD}$ viruses were generated similarly but with full-length FrMLV plasmid encoding Gag-Fluc and fusion-defective SFFV Gp55. ViralBoost reagent (ALSTEM, Richmond, CA) was added to producer cell plates 12 hr following transfection. 48 hr following transfection, culture supernatants were collected and filtered through 0.45 μm low protein-binding filters (Pall Corporation, Port Washington, NY). Viral stocks were aliquoted into 2 ml tubes and stored at –80°C.

## Virus titration

For titration of viral stocks, DFJ8 chicken fibroblasts (50,000 cells/well, 48-well plate; routinely checked for *Mycoplasma* contamination using MycoSensor Assay Kits, Agilent Inc) were infected with varying dilutions of viral stocks in the presence of 8 μg/ml polybrene (Sigma-Aldrich). 48 hr following infection, infected DFJ8 cells were analyzed by flow cytometry using antibodies to MLV GlycoGag. For preparation of single-cell suspensions, infected cells were incubated with Accutase (StemCell Technologies, Vancouver, Canada) for 5 min at 37°C. Accutase was neutralized via addition of RPMI containing 10% FBS. Cells were centrifuged for 5 min at ~110 × g and resuspended in PBS containing 1% BSA. Cells were fixed with 4% paraformaldehyde (PFA) for 7 min at room temperature. PFA was neutralized by addition of 0.1 M glycine in PBS. Cells were washed with PBS-0.1 M glycine twice and resuspended in FACS buffer solution (5% FBS, 1% BSA, 0.2% gelatin in PBS) for 15

min at room temperature. Cells were incubated with 1:500 anti-GlycoGag antibody conjugated with Alexa 647 at room temperature for 1 hr. Cells were washed twice with and resuspended in MACS Buffer (1X PBS, 2 mM EDTA, 1% BSA). Infectious units of fluorescent protein-encoding constructs were additionally enumerated by estimating the number of GFP or mCherry-expressing cells. Viral titers were determined by calculating infectious units per ml, based on numbers of infected cells resulting from each volume of virus supernatant used in titration.

## In vitro luciferase assays

Single-cell suspensions obtained from tissues or infected cells were lysed in 1X Passive Lysis Buffer (Promega) for 5 min at 37°C, and lysates were added to white-bottom 96-well flat-bottom plates (Costar, Corning, NY). Luciferase activity was measured after adding appropriate substrate (Promega Firefly Luciferase Assay substrate for firefly luciferase, or Promega nanoGlo nanoLuciferase substrate for NLuc and Antares, diluted 1:40 in PBS per manufacturer's instructions) using a Berthold luminometer (Berthold Technologies, Bad Wildbad, Germany). Luciferase activity associated with virions was performed on partially purified viruses that were sedimented through a 15% sucrose cushion in PBS at $25,000 \times g$ for 2 hr at 4°C. Sedimented viral pellets were resuspended in 0.1% BSA/PBS and diluted accordingly for detection within the luminometer linear range. For MLV Gag-Fluc virion luciferase assays, sedimented viral pellets were resuspended in 0.1% BSA/PBS or Passive Lysis Buffer (Promega). ATP-free or ATP-containing (150 µM ATP) substrate solutions were prepared containing 150 µg/ml D-luciferin (Goldbio, St. Louis, MO), 100 mM Tris pH 8, and 5 mM $MgCl_2$. Relative light units (RLU) were determined by taking luciferase readings of lysis buffer or PBS/0.1% BSA.

## Retrovirus administration

Virus stocks were stored at −80°C, thawed at 37°C, quickly placed on ice, and concentrated by sedimentation through a 15% sucrose-PBS cushion for 2 hr at 4°C, at $25,000 \times g$. After sedimentation, virus pellets were resuspended in endotoxin-free 0.1% BSA in PBS at appropriate luciferase light units or infectious units (IUs) for administration. Retroorbital (100 µL) and subcutaneous (intrafootpad or intramammary glands; 25 µl) injections were carried out using an insulin syringe with 31 G needle on anesthetized mice (0.5–5% isoflurane delivered using precision Dräger vaporizer with oxygen flow rate of 1 l/min). During retroorbital and subcutaneous virus inoculations, mice received $1 \times 10^5$ IU of MLV Env-Nluc for viral particle flow monitoring, $2 \times 10^4$ IU (corresponding to $2 \times 10^6$ RLU) of MLV Gag-Fluc $Env_{WT}$ or MLV Gag-Fluc $Env_{FD}$, for monitoring of cytoplasmic entry, or $5 \times 10^5$ IU of MLV 6ATRi-Nluc, MLV-Antares or WT MLV for monitoring of longitudinal virus spread and infectivity comparison. For mother-to-offspring transmission experiments, dams were inoculated with $1 \times 10^7$ IU of MLV Fluc-IRES-mCherry subcutaneously distributed into the mammary gland 6 days prior to transfer of neonates for fostering. For oral inoculation of neonatal mice, appropriate amounts of virus were resuspended in 15 µl sterile endotoxin-free PBS containing 0.1% BSA and 5% sucrose and fed using a p10 pipette tip. Mice were orally inoculated with $1 \times 10^6$ IU of MLV Env-Nluc for monitoring virus particle flow, $2 \times 10^4$ RLU of MLV Gag-Fluc for monitoring virus entry into cytoplasm, or $1 \times 10^6$ IU of MLV 6ATRi-Nluc, MLV pMIG-Nluc, or FrMLV WT for monitoring establishment of infection.

## Bioluminescence imaging (BLI)

### Image acquisition

All mice were anesthetized via isoflurane inhalation (3–5% isoflurane, oxygen flow rate of 1.5 l/min) prior and during BLI using the XGI-8 Gas Anesthesia System. Images were acquired with an IVIS Spectrum (PerkinElmer) and analyzed with the manufacturer's Living Image v4.7.3 in vivo software package. Image acquisition exposures were set to auto, with imaging parameter preferences set in order of exposure time, binning, and f/stop, respectively. Images were acquired with luminescent f/stop of 1, photographic f/stop of 8. Binning was set to medium.

### Short interval imaging for particle flow

Prior to imaging MLV Env-Nluc virion flow, mice received furimazine (Promega) diluted 1:40 in PBS. 100 µl diluted furimazine were administered retroorbitally. Image sequences were then acquired at

30 s intervals following administration of MLV Env-Nluc (r.o. or i.f.p). Image sequences were assembled and converted to videos using ImageJ.

### Imaging of adult mice

Mice infected with Antares- or Nluc-encoding viruses received 100 µl furimazine (NanoGlo furimazine, Promega) diluted 1:40 in sterile endotoxin-free PBS (r.o.) before imaging. Mice infected with Fluc-carrying viruses received in vivo grade D-luciferin (15 mg/ml in sterile endotoxin-free PBS, Gold Biotechnology Inc) (r.o.) before imaging.

### Imaging of neonatal mice

For non-invasive imaging of neonatal mice that were fed from infected dams, 25 µl of pre-warmed in vivo grade D-luciferin in PBS (15 mg/ml) was injected subcutaneously in the scruff of the neck. The substrate was allowed to diffuse for 10 min before imaging the mice using IVIS. For oral virus administration, neonatal mice were orally inoculated with various reporter viruses suspended in 15 µl of 5% sucrose in PBS. Luciferase-specific substrate was subcutaneously injected as above, 10 min before euthanasia. Infected areas of interest were identified by carrying out whole-body imaging following necropsy (*Ventura et al., 2019*). Infected regions indicated by luminescent signal were sampled and washed in PBS to remove residual blood and placed onto a glass plate. Additional droplets of furimazine in PBS (1:40) or D-luciferin (15 mg/ml) were added to organs and soaked in substrate for 1–2 min before BLI for quantitation.

### Image processing

Comparative images were compiled and batch-processed using the image browser with collective luminescent scales. Photon flux was measured as luminescent radiance (p/s/cm$^2$/sr). During luminescent threshold selection for image display, luminescent signals were regarded as background when minimum threshold levels resulted in displayed radiance above non-tissue-containing or known uninfected regions.

## Single-cell preparation from mouse tissue

pLNs, mSacs, and spleens harvested after necropsy were disrupted in serum-free media, treated with Liberase TL (0.2 mg/ml, Sigma-Aldrich) and DNase I (20 mg/ml, Roche) at 37°C for 20 min and passed through a 70 µm cell strainer (Falcon, Cat # 352350). Single-cell suspensions from footpad skin and tissue were isolated by first cutting into tiny pieces and treating with Liberase and DNAse I as above for 3 hr before passing them through a 100 µm strainer. Splenic cell suspensions were treated additionally with red blood cell lysis buffer at room temperature for 10 min (Sigma-Aldrich or BioLegend Inc) for removing RBCs to obtain single-cell suspensions. Single-cell suspensions from each lymphoid tissue or skin were fixed with 4% PFA (Electron Microscopy Sciences) before processing for flow cytometric analysis.

## Flow cytometry

Fixed single-cell suspensions were washed twice with 0.1 M glycine in PBS to neutralize excess PFA. Cells were blocked with CD16/32 antibodies (BioLegend Inc, San Diego, CA) in FACS staining buffer (5% FBS, 1% BSA, 0.2% Gelatin) or Cell staining buffer (BioLegend Inc, Cat # 420201) for 15 min to 1 hr. Infected cells were identified by staining them with Alexa Fluor 647 or Alexa Fluor 488 conjugated antibodies to Glycogag (mAb34 hybridoma). Hematopoietic cell types from footpads were identified using APC Rat anti-mouse CD45 (30-F11) (BD-Pharmingen, Cat # 559864), B cells were identified using PE/Cy7 anti-mouse CD19(6D5) (BioLegend, Cat # 115519, RRID:AB_313654), CD4$^+$ T cells were identified using PE/Cy7 anti-mouse CD4 (GK1.5) (BioLegend, Cat # 100421, RRID:AB_312706) and APC anti-mouse CD3ε Antibody (145-2 C11) (Cat # 100312, RRID:AB_312677), DCs were identified using AF647 anti-mouse CD11c (N418) (BioLegend, Cat # 117314, RRID:AB_492850) and PE anti-mouse/human CD11b (M1/70) (BioLegend, Cat # 101208, RRID:AB_312791). The cells were incubated with antibodies for 1–2 hr at room temperature. Flow cytometry was performed on a Becton Dickinson Accuri C6 benchtop cytometer. Data were analyzed with Accuri C6 or FlowJo v10 software (Treestar, Ashland, OR). 200,000–500,000 viable cells were acquired for each sample.

## Retrovirus challenge by ileal ligation

Ileal ligation was performed on anesthetized mice. Mice were first anesthetized with a ketamine/xylazine cocktail (ketamine 15 mg/ml xylazine 1 mg/ml) at 0.01 ml per g of body weight. Mice were then placed on isoflurane inhalation anesthesia using a Dräger vaporizer (1.5–2% isoflurane, flow rate 1 l/min). A small, ~5 mm incision was made above the abdominal cavity to expose the peritoneal muscle, in which a small ~3 mm incision was made above the intestine. The small intestine was carefully extracted using forceps. Small intestinal regions of interest were ligated by tying surgical suture (undyed, Vicryl braided P-3 polyglactin-coated J494G suture, Ethicon, Somerville, NJ) gently around serosal intestinal walls surrounding Peyer's Patches (PP). The suture needle was inserted through the mesenteric membrane, with care to avoid blood vessel obstruction. Knots were gently tied at each end of the PP, spanning a ~1–2 cm length. MLV Gag-GFP viruses corresponding to $1–4 \times 10^5$ IU based on comparative western blot analyses with antibodies to Gag with equivalent amounts of WT FrMLV were administered into the intestinal lumen through the intestinal wall using a 31 G needle. Total volume did not exceed 50 µl. Following the intestinal loop, excess suture was trimmed from knots and the intestine was carefully threaded back through the incision. The incision was sealed very gently using a small, low-tension binder clip, while the mouse remained under anesthesia until euthanasia at the end of the terminal surgical procedure. Intestinal loop inoculations did not exceed 1 hr. Anesthesia was closely monitored during the duration of the surgery and intestinal loop inoculation. Anesthetic planes were monitored every 15 min or more frequently based on heart rate, breathing rate and depth, and noxious stimuli reflexes as recommended by the Yale IACUC. At the end of 1 hr, the mouse was sacrificed, and the ligated intestinal region processed for cryosectioning and immunostaining.

## Cryo-immunohistology

Non-fluorescent-protein-containing tissue samples were harvested at indicated time points and fixed in 1× PBS containing freshly prepared 4% PFA for 1 hr at 4°C. Fluorescent-protein-containing samples were harvested and fixed in periodate-lysine-paraformaldehyde (PLP) fixative (1× PBS containing 1% PFA, 0.01 M sodium m-periodate and 0.075 M L-Lysine) for 30 min to 1 hr to preserve fluorescent protein fluorophores. Tissue samples were washed with PBS, dehydrated in a sucrose gradient consisting of 1 hr incubation at room temperature in 10, 20, or 30% sucrose in PBS, embedded and snap-frozen in Tissue-Tek O.C.T. compound (Sakura Finetek, Torrance, CA) and stored at –80°C. 15 µm tissue sections were cut on a Leica cryostat at −20°C and placed onto Superfrost Plus slides (Thermo Fisher). Tissue sections were dried at 37°C for 15 min and stored at −20°C for later use or rinsed in PBS for staining. Slides were washed in PBS (for stains of cell-surface proteins) or permeabilized with PBS containing 0.2% Triton X-100 (for stains of intracellular proteins) and treated with Fc receptor blocker (Innovex Biosciences, Richmond, CA) or Staining solution (5% FBS, 1% BSA, 0.2% gelatin) before staining with indicated antibodies in PBS containing 2% BSA. Staining was performed with the following antibodies: CD169-AF647, CD169-AF594, CD11c-eFluor450, CD11c-AF594, CD11c-AF647, and CD68-AF594 were from BioLegend Inc; GlycoGag-CF647 (mAb34 hybridoma, house-conjugated *Uchil et al., 2019b*; *Chesebro et al., 1981*). Stained sections were washed with PBS and a final rinse with water to minimize salt precipitation and mounted using ProLong Glass antifade reagent (Invitrogen, Thermo Fisher) and Fisher Finest thick coverslips (Thermo Fisher). Mounted slides were sealed with clear nail polish and cured for 1 hr or overnight at 37°C. Slides were analyzed by confocal microscopy using a Leica TCS SP8 microscope equipped with a white light and argon laser, and a Nikon W-1 Spinning Disk microscope. The images were processed using Volocity version 6.3 software (Perkin Elmer, Waltham, MA) and Nikon Elements software (Nikon, Tokyo, Japan). Figures were assembled with Photoshop CC and Illustrator CC (Adobe Systems, San Jose, CA).

## Electron microscopy and dual-axis tomography

Following removal or disarticulation, tissues were lightly fixed with 3% glutaraldehyde, 1% paraformaldehyde, and 5% sucrose in 0.1M sodium cacodylate trihydrate to render them safe from virus infectivity. Tissues were further dissected in cacodylate buffer, rinsed with cacodylate containing 10% Ficoll (70 kD, Sigma), which served as an extracellular cryoprotectant, placed in brass planchettes (Tell Pella, Inc, Redding, WA), and ultra-rapidly frozen with an HPM-010 high-pressure freezing

machine (BalTec/ABRA, Switzerland). Samples were then transferred under liquid nitrogen to cryo-tubes (Nunc) containing a frozen solution of 2% osmium tetroxide, 0.05% uranyl acetate in acetone, and placed in an AFS-2 freeze-substitution machine (Leica Microsystems, Wetzlar, Germany). Tissues were freeze-substituted at −90℃ for 72 hr, warmed to −20℃ over 12 hr, and held at that temperature for an additional 12 hr before warming to room temperature and infiltrating into Epon-Araldite resin (Electron Microscopy Sciences, Port Washington, PA). Samples were flat-embedded between two Teflon-coated glass microscope slides and the resin polymerized at 60℃ for 48 hr. Embedded tissue blocks were observed by light microscopy to ascertain preservation quality and select gross regions of interest. Blocks were extracted with a scalpel and glued to plastic sectioning stubs prior to sectioning. Semi-thick (300–400 nm) sections were cut with a UC6 ultramicrotome (Leica Microsystems) using a diamond knife (Diatome, Ltd., Nidau, Switzerland) Sections were placed on formvar-coated copper-rhodium slot grids (Electron Microscopy Sciences) and stained with 3% uranyl acetate and lead citrate. Colloidal gold particles (10 nm) were placed on both surfaces of the grids to serve as fiducial markers for tomographic image alignment. Grids were placed in a dual-axis tomography holder (Model 2010, E.A. Fischione Instruments, Export PA) and imaged with a Tecnai TF30ST-FEG transmission electron microscope (300 keV; Thermo Fisher Scientific). Images were recorded with a 2k × 2k CCD camera (XP1000; Gatan, Pleasonton, CA). Tomographic tilt series and large-area montages were acquired automatically using the SerialEM software package (*Beutner et al., 1994*). For dual-axis tomography, images were collected at 1˚ intervals as samples were tilted ±64˚. The grid was then rotated 90˚ and a second tilt-series was acquired about the orthogonal axis. Tomograms were calculated, analyzed, and modeled using the IMOD software package (*Held et al., 1993*; *Oetke et al., 2006*) on MacPro and iMac Pro computers (Apple, Inc, Cupertino, CA). Cell types and frequency within tissue sections were identified using 2D montaged overviews. Virus particles and infected cells were further characterized in 3D by high-resolution electron tomography.

## In vivo nanoluc reporter stability assays

In vitro stability of a full-length MLV construct similar to the one generated by us in this study carrying IRES-driven nLuc reporter has been carried out previously (*Logg et al., 2001*). As our current study has an in vivo focus, we sought to determine the reporter stability under these conditions. For determining in vivo longitudinal stability of Nluc in replication-competent reporter virus, B6 mice (n = 5) were challenged retroorbitally with MLV 6ATRi-Nluc ($2 \times 10^6$ IU). RNA was extracted using Qiagen total RNA extraction kit from 100 µl blood drawn longitudinally (3, 7, 10, 14 dpi) or from splenocytes (14 dpi) after necropsy. RNA was first converted to cDNA using iScript cDNA Synthesis Kit (Bio-Rad). cDNAs were used for RT$^2$-qPCR analyses using PowerUp SYBR Green Master Mix using primers for Nluc (nluc F- GGAGGTGTGTCCAGTTTGTT nluc R- ATGTCGATCTTCAGCCCATTT) and MLV gag (gag F –AAGGCGCTGGGTTACATTCT gag R - GAGAGAGGGGAGGTTTAGGGT). Stability of Nluc within the viral genome was estimated by determining the ratios of threshold cycle ($C_t$) values for MLV gag and Nluc. RNA isolated from cultured DFJ8 cells infected with MLV 6ATRi-Nluc (36 hpi) were used as reference standard for Gag:Nluc ratio. Uninfected DFJ8 cells were used as negative controls to ascertain that the primers do not amplify non-specific products.

## Statistical analyses

Statistical comparisons were performed using non-parametric Mann–Whitney tests (two-tailed) available in GraphPad Prism software (GraphPad Software, La Jolla, CA). A difference was considered significant if $p < 0.05$.

## Acknowledgements

This work was supported by NIH grants R01 CA098727 to WM and P50AI150464 to WM and PJB; R33AI122384 and R01AI145164 to PK, the Flow Cytometry Shared Resource of the Yale Cancer Center P30 CA016359, Yale Center for Cellular and Molecular Imaging S10 OD020142, Virology Training Grant fellowship T32AI055403 to KAH, the Gruber Foundation to MWG, and a fellowship from the China Scholarship Council – Yale World Scholars to RP. We thank the Kavli Nanoscience Institute at Caltech for maintenance of the TF-30 electron microscope.

## Additional information

### Competing interests

Pamela J Bjorkman: Reviewing editor, *eLife*. The other authors declare that no competing interests exist.

### Funding

| Funder | Grant reference number | Author |
| --- | --- | --- |
| National Cancer Institute | R01 CA098727 | Walther Mothes |
| National Institute of Allergy and Infectious Diseases | 5P50AI150464-14 | Pamela J Bjorkman Walther Mothes |
| National Institute of Allergy and Infectious Diseases | 5R33AI122384-05 | Priti Kumar |
| National Institute of Allergy and Infectious Diseases | 5R01AI145164-03 | Priti Kumar |
| National Institute of Allergy and Infectious Diseases | T32AI055403 | Kelsey A Haugh |
| Gruber Foundation | | Michael W Grunst |
| China Scholarship Council | China Scholarship Council – Yale World Scholars Program | Ruoxi Pi |

The funders had no role in study design, data collection and interpretation, or the decision to submit the work for publication.

### Author contributions

Kelsey A Haugh, Conceptualization, Data curation, Formal analysis, Validation, Investigation, Visualization, Methodology, Writing - original draft, Writing - review and editing; Mark S Ladinsky, Conceptualization, Data curation, Formal analysis, Validation, Investigation, Methodology, Writing - review and editing; Irfan Ullah, Data curation, Formal analysis, Investigation; Helen M Stone, Alexandre Gilardet, Investigation; Ruoxi Pi, Formal analysis, Investigation, Methodology, Writing - review and editing; Michael W Grunst, Investigation, Writing - review and editing; Priti Kumar, Resources, Funding acquisition, Project administration; Pamela J Bjorkman, Resources, Methodology, Project administration; Walther Mothes, Conceptualization, Resources, Supervision, Funding acquisition, Visualization, Methodology, Writing - original draft, Project administration, Writing - review and editing, co-corresponding author; Pradeep D Uchil, Conceptualization, Formal analysis, Supervision, Validation, Investigation, Visualization, Methodology, Writing - original draft, Project administration, Writing - review and editing

### Author ORCIDs

Kelsey A Haugh https://orcid.org/0000-0002-4277-2493
Mark S Ladinsky http://orcid.org/0000-0002-1036-3513
Pamela J Bjorkman http://orcid.org/0000-0002-2277-3990
Walther Mothes http://orcid.org/0000-0002-3367-7240
Pradeep D Uchil https://orcid.org/0000-0002-7236-858X

### Ethics

Animal experimentation: All experiments were approved by the Institutional Animal Care and Use Committees (IACUC) protocols 2020-10649 and Institutional Biosafety Committee of Yale University (IBSCYU). All the animals were housed under specific pathogen-free conditions in the facilities provided and supported by Yale Animal Resources Center (YARC). All IVIS imaging, blood draw and virus inoculation experiments were done under anesthesia using regulated flow of isoflurane:oxygen mix to minimize pain and discomfort to the animals. Animals were housed under specific pathogen-free conditions in the Yale Animal Resources Center (YARC) in the same room of the vivarium. Yale

University is registered as a research facility with the United States Department of Agriculture, License and Registration number 16-R-0001 Registered until March 20, 2023. It also is fully accredited by the Association for Assessment and Accreditation of Laboratory Animal Care (AAALAC) AAALAC Accreditation: April 3, 2019. An Animal Welfare Assurance (#D16-0014) is on file with OLAW-NIH; effective May 1, 2019-May 31, 2023.

## Decision letter and Author response
Decision letter https://doi.org/10.7554/eLife.64179.sa1
Author response https://doi.org/10.7554/eLife.64179.sa2

# Additional files
## Supplementary files
• Transparent reporting form

## Data availability
Data is plotted as individual points wherever possible. We can provide Graphpad prism files that was used to plot all the graphs for each figure upon request. Raw datasets are freely available upon request. Interested parties should contact https://medicine.yale.edu/profile/pradeep_uchil/, https://medicine.yale.edu/profile/walther_mothes/, and we will place requested dataset onto an externally accessible Yale Box Server. Requestors will then be provided with a direct URL link from which they can download the files at their convenience. All the images acquired using confocal microscopy are available at Dryad https://doi.org/10.5061/dryad.hhmgqnkgw.

The following dataset was generated:

| Author(s) | Year | Dataset title | Dataset URL | Database and Identifier |
|---|---|---|---|---|
| Haugh KA, Ladinsky MS, Ullah I, Stone HM, Pi R, Gilardet A, Grunst MW, Kumar P, Bjorkman PJ, Mothes W, Uchil PD | 2021 | In vivo imaging of retrovirus infection reveals a role for Siglec-1/CD169 in multiple routes of transmission | https://doi.org/10.5061/dryad.hhmgqnkgw | Dryad Digital Repository, 10.5061/dryad.hhmgqnkgw |

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
