## [Decision Letter]

**Acceptance summary:**

The manuscript describes the use of sophisticated imaging techniques, pioneered by this group in previous studies, to track the early steps of murine leukemia virus (MLV) infection in mice. The study extends previous work on i.v. and s.c. infection, and the key role of Siglec-1/CD169 and sentinel macrophages, by showing that similar processes apply to infection by milk-borne transmission in the GI tract.

**Decision letter after peer review:**

Thank you for submitting your article "in vivo imaging of retrovirus infection reveals a role for Siglec-1/CD169 in multiple routes of transmission" for consideration by *eLife*. Your article has been reviewed by 3 peer reviewers, and the evaluation has been overseen by a Reviewing Editor and Sara Sawyer as the Senior Editor. The reviewers have opted to remain anonymous.

The manuscript describes the use of sophisticated imaging techniques, pioneered by this group in previous studies, to track the early steps of murine leukemia virus (MLV) infection in mice. The study extends previous work on i.v. and s.c. dissemination and the key role of Siglec 1/CD169) and, importantly, show that similar processes apply to infection via an oral route.

Essential Revisions:

1) While expressing enthusiasm for the study, the reviewers indicate that the impact of the work would be raised significantly through the addition of information on the cell types infected by MLV following its transfer to lymphoid tissues.

2) The reviewers raise a number of other issues that you should consider and address either through modifications to the text or through the inclusion of additional data.

*Reviewer #2:*

1. The manuscript would give much greater insight into how infection occurs if there was analysis of the cell types that were infected in Figures 2E, G, H; 3F, G; 6F, G and 9E, at minimum. While the localization of virus in organs is interesting, it does not really provide insight into how retrovirus infection is established in vivo.

2. Lines 134-6. Discussion of the gradual decline in signal. Do the authors envision that infected cells are being eliminated by the immune system? Can the authors do simultaneous PCR to show that the cells are really being eliminated, rather than just losing signal.

3. Lines 165-7. Do the authors think that macrophage capture of virus is occurring within 3 min?

4. Figure 3C – Why does capture by CD169+ macrophages require a fusion-competent envelope? The difference in virus levels at different time points, particularly 6 hr, is not very convincing. It looks like the only reason there is a difference at later time points is that the levels of EnvFD-containing virus decrease. If one just compares MLV Gag-Fluc Env WT virus levels over time, are they statistically different? Why do EnvFD virus levels decrease with time?

5. Lines 203 -6. Why isn't there immune control of infection in the mammary gland?

6. Figure 5A. What cell types in the mammary gland are getting infected? I'm surprised that "infection" can really be established in adult females in such a short infection time period and during lactation, when the mammary epithelia are no longer replicating, and to sufficient levels so that virus can be shed and detected in milk. Was this a single experiment?

7. Same question as for 3C – why is a fusion-competent virus required if its CD169+ macrophage capture of virus?

8. Figure 8. Does capture by the Peyer's patch-resident macrophages require fusion-competent MLV? In the low magnification photo in Figure 8b, it looked like there was also a lot of virus penetration into areas not populated by CD169+ macrophages. How are the authors sure that this was specific capture?

*Reviewer #3:*

1) The design of the various reporter viruses is instrumental for this study but to make the results obtained with these viruses comparable, additional side-by-side characterization (e.g. relative infectivity, labelling efficacies) would be helpful. For replication competent reporters it will be important to document how stable the reporter is over multiple rounds of infection in vitro and in vivo.

2) Figure 2A displays prominent fluorescence in liver and spleen but in early time points, a third spot can be seen – what is this and is this tropism relevant? How well does furimazine distribute to different tissues and is this a relevant factor in where and with which sensitivity virus can be detected?

3) Immune responses are mentioned several times in the manuscript but were not analyzed. Do the authors have data or could discuss whether e.g. the accumulation of trapped particles increases innate immune recognition and if depletion of CD169+ macrophages affects immune responses beyond their role in virus capture? Figure 3H is mentioned in the text but not shown.

4) For the experiments using subcutaneous challenges, CD169-/- animals need to be included to quantify the impact the impact of CD169+ macrophages to virus capture (as was done in Figure 9 for oral challenge).

5) After oral challenge, the lack of CD169 prevents particle accumulation in PP and flow to mSac, but residual infection of mSac is maintained (Figure 9). Please comment/discuss. Does CD169 ko affect mSac morphology?

---

## [Author Response]

Essential Revisions:1) While expressing enthusiasm for the study, the reviewers indicate that the impact of the work would be raised significantly through the addition of information on the cell types infected by MLV following its transfer to lymphoid tissues.

We thank the reviewers for their critique to help improve our manuscript. We have now included data for the cell types infected at the footpad during subcutaneous challenge and in the mesenteric sac (mSac) during oral transmission. We have referenced our previous studies where we had characterized MLV-infected cells in-depth in the popliteal lymph nodes and spleen.

2) The reviewers raise a number of other issues that you should consider and address either through modifications to the text or through the inclusion of additional data.

Please find our point-by-point response to the reviewer concerns below.

Reviewer #2:1. The manuscript would give much greater insight into how infection occurs if there was analysis of the cell types that were infected in Figures 2E, G, H; 3F, G; 6F, G and 9E, at minimum. While the localization of virus in organs is interesting, it does not really provide insight into how retrovirus infection is established in vivo.

We have characterized the infected cells in-depth in the pLN (s.c. route; B-1a, naïve B-2, CD4^+^ T cells) and spleen (retroorbital route; predominantly follicular, marginal zone, and transitional B cells) in our previous publications [7], [8], respectively. These details and references have now been added. We had however not characterized infected cells in the footpad during s.c challenge and in the mesenteric sacs during oral challenge. We have now included a profile of infected cells in both footpad and mSac as per reviewer request. Specifically, we found that DCs (CD11b^+^CD11c^+^ and CD11c^+^) are infected in footpads (Figure 3 —figure supplement 1). In mSacs, DCs, B cells and CD4^+^ T cells are targeted by MLV for establishment of infection (Figure 6 —figure supplement 1).

2. Lines 134-6. Discussion of the gradual decline in signal. Do the authors envision that infected cells are being eliminated by the immune system? Can the authors do simultaneous PCR to show that the cells are really being eliminated, rather than just losing signal.

We have characterized temporal levels of infection in B6 and CD169^-/-^ mice with WT type FrMLV using antibodies to Gag followed by FACS analyses to identify the number of infected cells in our previous publications [7, 8]. The data indeed point to an elimination of infected cells in adult B6 mice by the immune system. We now refer to them in the manuscript for clarity (Line 138-145).

3. Lines 165-7. Do the authors think that macrophage capture of virus is occurring within 3 min?

The signal near the footpad injection site is a combination of virus capture by resident tissue architecture like collagen fibrils, muscle, and cell types like macrophages and DCs. In our previous study, we have shown by intravital imaging that lymph-borne retroviruses arriving at the draining pLN are captured instantaneously by SCS CD169 macrophages [7].

4. Figure 3C – Why does capture by CD169+ macrophages require a fusion-competent envelope?

CD169 capture does not require fusion competent envelope. Viruses will be captured by CD169 macrophages even if the virus lacks envelope protein as capture is dependent on gangliosides present on viral membranes. It is fusion into target cells and detection of cytoplasmic Fluc activity that requires a fusion competent envelope.

The difference in virus levels at different time points, particularly 6 hr, is not very convincing. It looks like the only reason there is a difference at later time points is that the levels of EnvFD-containing virus decrease. If one just compares MLV Gag-Fluc Env WT virus levels over time, are they statistically different? Why do EnvFD virus levels decrease with time?

Ideally the EnvFD virus should not produce any Fluc signal as Fluc is enclosed within virion membranes. The resident signal that we see is due to presence of lysed virions that arise during the process of virus purification or during virus administration.

As the previous data was not statistically robust, we have repeated this experiment to increase the number of mice (n = 8- 10) and lengthened the time span of the experiment to 72 h. These data are now included in the manuscript along with statistical analyses in Figure 3.

5. Lines 203 -6. Why isn't there immune control of infection in the mammary gland?

There is immune control of virus in lactating dams and luciferase signal increase and go down with time. However, we find that the FrMLV infection in individual mammary gland peak at different time points based on initial level of infection. These data are now presented in Figure 5A as well as figure supplement 1 for Figure 5.

6. Figure 5A. What cell types in the mammary gland are getting infected? I'm surprised that "infection" can really be established in adult females in such a short infection time period and during lactation, when the mammary epithelia are no longer replicating, and to sufficient levels so that virus can be shed and detected in milk. Was this a single experiment?

We have performed this experiment several times and is reproducible. We now include data documenting the increase in signal indicating that the virus is indeed replicating in mammary glands (see previous comment). To demonstrate that this is reproducible we have included data from 3 mice (Figure 5A and figure supplement 1 for Figure 5). Interestingly as noted in previous comment, the progression of infection in mammary glands are not synchronous. This may be due to different levels of cell division in mammary glands during lactation. We also provide immunofluorescence data in Figure 5 that show FrMLV-infected epithelial cells in mammary glands identified by bioluminescence.

7. Same question as for 3C – why is a fusion-competent virus required if its CD169+ macrophage capture of virus?

CD169 capture does not require fusion competent envelope. Viruses will be captured by CD169 macrophages even if the virus lacks envelope protein, as capture is dependent on gangliosides present on viral membranes. It is fusion into target cells that require fusion competent envelope. CD169 macrophages do not get infected. Rather they promote infection of susceptible lymphocytes by a process called trans-infection.

8. Figure 8. Does capture by the Peyer's patch-resident macrophages require fusion-competent MLV? In the low magnification photo in Figure 8b, it looked like there was also a lot of virus penetration into areas not populated by CD169+ macrophages. How are the authors sure that this was specific capture?

Fusion-competent viruses are not required to reach the Peyer patch and even mSacs. Labeled virus particles alone arrive at Peyer’s patches and mSac. In addition, virus enter into Peyer's patch even in CD169^-/-^ mice, accounting for the virus particles observed in this image. With respect to specific capture by CD169, our data using KO mice demonstrate that CD169 contributes to virus acquisition in neonatal mice that were challenged orally.

Reviewer #3:1) The design of the various reporter viruses is instrumental for this study but to make the results obtained with these viruses comparable, additional side-by-side characterization (e.g. relative infectivity, labelling efficacies) would be helpful. For replication competent reporters it will be important to document how stable the reporter is over multiple rounds of infection in vitro and in vivo.

We now provide the data for longitudinal in vivo stability of the Nluc reporter in the revised manuscript and find that Nluc is stable till 14 days post infection (Figure 2I). In vitro stability for a construct like ours were reported previously [9]. We now refer to this report in Materials section.

2) Figure 2A displays prominent fluorescence in liver and spleen but in early time points, a third spot can be seen – what is this and is this tropism relevant?

Liver has four major lobes and, in the image, the third spot is also liver.

How well does furimazine distribute to different tissues and is this a relevant factor in where and with which sensitivity virus can be detected?

Bioavailability of furimazine is high throughout the organs and it can also diffuse into brain by bypassing blood brain barrier and immunologically privileged organs like testes. Sensitivity also depends on the depth of the tissue. Hence in many of our experiments where signal is not detected non-invasively, we have resorted to imaging the organs after necropsy to improve sensitivity.

3) Immune responses are mentioned several times in the manuscript but were not analyzed. Do the authors have data or could discuss whether e.g. the accumulation of trapped particles increases innate immune recognition and if depletion of CD169+ macrophages affects immune responses beyond their role in virus capture?

We have carried out in-depth studies to look at immune responses (innate, humoral and cell mediated) in B6 vs CD169^-/-^ mice upon retrovirus infection in our two previous studies [8, 10]. We have referred to these studies in the revised manuscript for better reader comprehension.

Figure 3H is mentioned in the text but not shown.

We apologize for the error. This should be Figure 3F.

4) For the experiments using subcutaneous challenges, CD169-/- animals need to be included to quantify the impact the impact of CD169+ macrophages to virus capture (as was done in Figure 9 for oral challenge).

We have carried out in-depth comparative studies to look at capture and infection in B6 and CD169 KO mice in our three previous studies [7, 8, 10]. We have referred to these studies in the text now.

5) After oral challenge, the lack of CD169 prevents particle accumulation in PP and flow to mSac, but residual infection of mSac is maintained (Figure 9). Please comment/discuss. Does CD169 ko affect mSac morphology?

CD169 KO does not have any effect on lymph node and mSac morphology as the macrophages are intact only the CD169 gene is deleted. CD169 promotes infection by capturing lymph and blood-borne virus. In its absence the infection is diminished but not eliminated [7].

References:

1. Moseman EA, Iannacone M, Bosurgi L, Tonti E, Chevrier N, Tumanov A, et al. B cell maintenance of subcapsular sinus macrophages protects against a fatal viral infection independent of adaptive immunity. Immunity. 2012;36(3):415-26. doi: 10.1016/j.immuni.2012.01.013. PubMed PMID: 22386268; PubMed Central PMCID: PMC3359130.

2. Iannacone M, Moseman EA, Tonti E, Bosurgi L, Junt T, Henrickson SE, et al. Subcapsular sinus macrophages prevent CNS invasion on peripheral infection with a neurotropic virus. Nature. 2010;465(7301):1079-83. doi: 10.1038/nature09118. PubMed PMID: 20577213; PubMed Central PMCID: PMCPMC2892812.

3. Junt T, Moseman EA, Iannacone M, Massberg S, Lang PA, Boes M, et al. Subcapsular sinus macrophages in lymph nodes clear lymph-borne viruses and present them to antiviral B cells. Nature. 2007;450(7166):110-4. doi: 10.1038/nature06287. PubMed PMID: 17934446.

4. Gonzalez SF, Lukacs-Kornek V, Kuligowski MP, Pitcher LA, Degn SE, Kim YA, et al. Capture of influenza by medullary dendritic cells via SIGN-R1 is essential for humoral immunity in draining lymph nodes. Nat Immunol. 2010;11(5):427-34. doi: 10.1038/ni.1856. PubMed PMID: 20305659; PubMed Central PMCID: PMC3424101.

5. Chatziandreou N, Farsakoglu Y, Palomino-Segura M, D'Antuono R, Pizzagalli DU, Sallusto F, et al. Macrophage Death following Influenza Vaccination Initiates the Inflammatory Response that Promotes Dendritic Cell Function in the Draining Lymph Node. Cell Rep. 2017;18(10):2427-40. doi: 10.1016/j.celrep.2017.02.026. PubMed PMID: 28273457.

6. Heesters BA, Chatterjee P, Kim YA, Gonzalez SF, Kuligowski MP, Kirchhausen T, et al. Endocytosis and recycling of immune complexes by follicular dendritic cells enhances B cell antigen binding and activation. Immunity. 2013;38(6):1164-75. doi: 10.1016/j.immuni.2013.02.023. PubMed PMID: 23770227; PubMed Central PMCID: PMC3773956.

7. Sewald X, Ladinsky MS, Uchil PD, Beloor J, π R, Herrmann C, et al. Retroviruses use CD169-mediated trans-infection of permissive lymphocytes to establish infection. Science. 2015;350(6260):563-7. doi: 10.1126/science.aab2749. PubMed PMID: 26429886; PubMed Central PMCID: PMCPMC4651917.

8. Uchil PD, π R, Haugh KA, Ladinsky MS, Ventura JD, Barrett BS, et al. A Protective Role for the Lectin CD169/Siglec-1 against a Pathogenic Murine Retrovirus. Cell Host Microbe. 2019;25(1):87-100 e10. doi: 10.1016/j.chom.2018.11.011. PubMed PMID: 30595553; PubMed Central PMCID: PMC6331384.

9. Logg CR, Logg A, Tai CK, Cannon PM, Kasahara N. Genomic stability of murine leukemia viruses containing insertions at the Env-3' untranslated region boundary. J Virol. 2001;75(15):6989-98. doi: 10.1128/JVI.75.15.6989-6998.2001. PubMed PMID: 11435579; PubMed Central PMCID: PMCPMC114427.

10. Pi R, Iwasaki A, Sewald X, Mothes W, Uchil PD. Murine Leukemia Virus Exploits Innate Sensing by Toll-Like Receptor 7 in B-1 Cells To Establish Infection and Locally Spread in Mice. J Virol. 2019;93(21). doi: 10.1128/JVI.00930-19. PubMed PMID: 31434732; PubMed Central PMCID: PMC6803250.